



# ROMSOC: A regional atmosphere-ocean coupled model for CPU-GPU hybrid system architectures

Gesa K. Eirund[1], Matthieu Leclair[2], Matthias Muennich[1], and Nicolas Gruber[1,2]

[1]Environmental Physics, Institute of Biogeochemistry and Pollutant Dynamics, ETH Zurich
[2]Center for Climate Systems Modeling (C2SM), ETH Zurich

**Correspondence:** Gesa K. Eirund (gesa.eirund@alumni.ethz.ch)

**Abstract.**

Recent years have seen significant efforts to refine the horizontal resolutions of global and regional climate models to the kilometer scale. This refinement aims to better resolve atmospheric and oceanic mesoscale processes, thereby improving the fidelity of simulations. However, these high-resolution simulations are computationally demanding, often necessitating trade-

offs between resolution and simulated timescale. A key challenge is that many existing models are designed to run on central processing units (CPUs) alone, limiting their ability to leverage the full computational power of modern supercomputers, which feature hybrid architectures with both CPUs and graphics processing units (GPUs).

In this study, we introduce ROMSOC, a newly developed regional coupled atmosphere-ocean model. ROMSOC integrates the Regional Oceanic Modeling System (ROMS) in its original CPU-based configuration with the Consortium for Small-

Scale Modeling (COSMO) model (v5.12), which can utilize GPU accelerators on heterogeneous system architectures. This combination efficiently exploits the hybrid CPU-GPU architecture of the Piz Daint supercomputer at the Swiss National Supercomputing Centre (CSCS), achieving a speed-up of up to six times compared to a CPU-only version with the same number of nodes.

We evaluated the model using a configuration focused on the northeast Pacific, where ROMS covers the entire Pacific Ocean

with a telescopic grid, providing full ocean mesoscale-resolving refinement in the California Current System (CalCS; 4 km resolution). Meanwhile, COSMO covers most of the northeast Pacific at a 7 km resolution. This configuration was run in hindcast mode for the years 2010-2021, examining the roles of different modes of air-sea coupling at the mesoscale, including thermodynamical coupling (associated with heat fluxes) and mechanical coupling (associated with wind stress and surface ocean currents).

Our evaluation indicates that the hindcast generally agrees well with observations and reanalyses. Notably, large-scale sea surface temperature (SST) patterns and coastal upwelling are well-represented, but SSTs show a small cold bias, resulting from too-strong wind forcing. Additionally, the coupled model exhibits a deeper and more realistic simulation of the ocean mixed-layer depth with a more pronounced seasonal cycle, driven by the enhanced wind-driven mixing. On the other hand, our ROMSOC simulations reveal a negative cloud cover bias off the coast of southern California, a common issue in climate

models.



# 1 Introduction

Small-scale processes in the atmosphere and ocean, such as shallow and deep atmospheric convection (Prein et al., 2015), clouds (Schneider et al., 2017), interactions of large-scale flow with topography (Small et al., 2015), or ocean mesoscale variability (McClean et al., 2011) require model resolutions approaching scales of a few kilometers or finer, in order to be explicitly computed. These small-scale processes are not only essential for synoptic weather variability and extremes, but also determine essential climate properties such as climate sensitivity and hence the level of projected climate change (Palmer, 2014; Palmer and Stevens, 2019; Kirtman et al., 2012; Meredith et al., 2015; Stevens et al., 2020; Caldwell et al., 2021). However, the advantage of higher resolution comes at the cost of shorter simulation periods or smaller domain sizes. This compromise results from limited available computational power as well as the often neglected large energy footprint of computationally heavy calculations, which results in high operational costs of supercomputing centers and large environmental footprints (Jones, 2018).

To meet these shortcomings, the most advanced supercomputing systems now employ hybrid architectures comprising both central processing units (CPUs) and graphics processing units (GPUs), which can achieve a substantially faster data treatment through a higher degree of parallel computing. Yet, nearly all currently existing weather and climate models were developed to run on CPU architectures. In order for these model codes to be able to run on GPUs, they have to be adapted and rewritten, which requires a tremendous amount of effort. Yet, a few attempts have been made to port models to GPU architectures, and several more are currently underway: this includes NICAM (Nonhydrostatic ICosahedral Atmospheric Model; Demeshko et al., 2013), COSMO (Consortium for Small-Scale Modeling; Fuhrer et al., 2018), LICOM3 (LASG/IAP Climate System Ocean Model version 3; Wang et al., 2021) and ICON (Icosahedral Nonhydrostatic Weather and Climate Model; Giorgetta et al., 2022). With these advances, it has been possible to refine the horizontal resolutions of climate models down to the kilometer-scale, while keeping large domain sizes and multiyear-long simulation time scales (Schär et al., 2020; Stevens et al., 2020). This allowed these models to resolve small-scale processes such as atmospheric convection (Leutwyler et al., 2016), cloud dynamics (Heim et al., 2021), or stratospheric dynamics such as the gravity wave-driven Quasi-biennial oscillation (Giorgetta et al., 2022).

Many of these models consider only the atmosphere and use observed or constant sea surface temperatures (SSTs) as lower boundary condition. As the weather and climate system is inherently an atmosphere-ocean coupled system and many weather/climate phenomena (e.g., extreme events or hurricanes) are driven by both atmospheric and oceanic weather, coupled atmosphere-ocean models at high resolution are needed to simulate these processes. Mauritsen et al. (2022) and Hohenegger et al. (2023) presented such high-resolution coupled simulations from the global ICON-Sapphire experiments at a grid spacing of 5 km. Similarly, Takasuka and Satoh (2024) target to evaluate a one-year long simulation of several GSRMs (Global Storm Resolving Models) and present results for the NICAM model on a global 3.5 km mesh. However, given the high computational





demand, these simulations were limited to cover a year to a decade of simulation time.

Longer coupled simulations are possible using regional climate models, which are limited by area but less by time. Regional climate models have the advantage of providing high-resolution climate projections, when driven by a future scenario of a GCM on their lateral boundaries (so-called dynamic downscaling). They are able to capture local climate phenomena and hence contribute to a better understanding of regional climate dynamics.

For these regional models, commonly used atmospheric models like the Weather Research Forecast (WRF, Skamarock
et al., 2021) model or the COSMO model (Schättler et al., 2000) are coupled to regional oceanic modeling systems such as the Regional Oceanic Modeling System (ROMS) (Shchepetkin and McWilliams, 2005), the Nucleus for European Modelling of the Ocean (NEMO, Gurvan; et al., 2016) model, or the Coastal and Regional Ocean Community Model (CROCO, Auclair; et al.).

One example of a coupled system set up for the Southern Ocean was presented by Byrne et al. (2016) who performed coupled
simulations using COSMO-ROMS, with a uniform horizontal resolution of 10 km for the atmosphere and ocean and covering the same grid domains. However, these simulations represented primarily idealized experiments and lasted for a few months only. On the more climatological scale, Renault et al. (2020) performed a 16-year hindcast simulation for the California Current System employing WRF-ROMS with a horizontal resolution of 6 and 4 km for the atmosphere and ocean, respectively, for the California Current System (CalCS). The coupled model compared well with observations, but its domain was limited to a 32
x 28° box in the longitudinal x latitudinal direction off California, thus not covering any basin-wide coastal and open-ocean processes, which can influence local dynamics (Frischknecht et al., 2015).

To overcome these shortcomings, we present an atmosphere-ocean coupled model configuration for regional scales, which comprises ROMS as the oceanic component coupled to COSMO, building on previous work by Byrne et al. (2016). Our coupled model system is set up for unequal grid spacings and domains in the CalCS, thus resolving both local, as well as remote
dynamical forcings. As the CalCS represents a region of particular importance with respect to atmosphere-ocean interactions, we focus on this particular region with our modeling efforts. There, any atmosphere-ocean interactions also have the potential to affect biogeochemistry, and hence ocean productivity and marine life (Gruber et al., 2011), adding another dimension to the coupled climate system. In the atmosphere along the California coast, low clouds, which in this region are primarily marine stratocumulus, are a common feature, capped by a strong inversion layer that forms through the interaction between descending
air within the high pressure system and the cool marine boundary layer driven by the coastal upwelling of cold water along the US west coast (Klein and Hartmann, 1993). Marine stratocumulus clouds play a crucial role determining the surface energy budget. Due to their high albedo, they reflect around 35-42% of the incoming solar radiation (Bender et al., 2011) and hence determine the amount of solar radiation available for surface heating (and hence thermal ocean mixing) as well as ocean productivity. In turn, the ocean impacts their occurrence through turbulent heat fluxes and the distribution of SST patterns.
Despite their importance for global and regional climate, global models still exhibit large biases in their representation (e.g., Teixeira et al., 2011). In addition, in global climate models, eastern boundary systems, such as the CalCS, are characterized by large SST biases (up to +3 °C, Richter, 2015), likely due to the model's coarse resolution and a resulting misrepresentation of





coastal upwelling, cloud cover, surface wind patterns, and the cross-shore eddy heat flux. Thus, we expect our high-resolution regional coupled model to show smaller biases in the upwelling region as a result of the high resolution atmospheric forcing.


In order to increase performance for our computationally expensive simulations, we use the GPU-enabled version of COSMO (v5.12), while keeping ROMS in its original configuration for CPU. This permits us to make efficient use of the hybrid system architecture of the Piz Daint supercomputer at CSCS.

We present this novel model configuration in section 2, while in section 3 we address the model performance. We then

evaluate our coupled model for the atmosphere and the ocean (section 4) and finish the paper with the a discussion (section 5) and conclusions and outlook (section 6).

## 2  Model configuration

### 2.1  The regional coupled model ROMSOC

Our coupled model comprises of ROMS for the ocean and COSMO for the atmosphere, respectively, and is configured for the

Northeast Pacific with a focus on the CalCS.

ROMS is a three-dimensional ocean general circulation model, which is used here in its UCLA-ETH version (Marchesiello et al., 2003; Frischknecht et al., 2015). It computes the hydrostatic primitive equations of flow for the evolution of the prognostic variables potential temperature and salinity, surface elevation, and the horizontal velocity components (Shchepetkin and McWilliams, 2005). Horizontally, the model grid uses curvilinear coordinates while vertically, terrain-following coordinates

including a time-varying free surface are employed. Higher vertical resolutions at the ocean surface and bottom allow for an appropriate representation of the oceanic boundary layer (Marchesiello et al., 2003; Gruber et al., 2006). Vertical mixing is parameterized using the first-order K-profile boundary layer scheme by Large et al. (1994). In addition, ROMS includes the Biogeochemical Elemental Cycling (BEC) model that describes the functioning of the lower trophic ecosystem in the ocean and the associated biogeochemical cycles (Frischknecht et al., 2018; Desmet et al., 2022; Koehn et al., 2022). This additional

coupling to the BEC model allows for analyses of the biogeochemical response to atmospheric coupling, such as the locations of nutrient-rich waters and ocean productivity changes, which is another unique feature of our coupled model.

COSMO is a nonhydrostatic, limited-area atmosphere model, which is run in Climate Limited-Area Modeling (CLM) mode to allow for longer-term simulations. We use a refactored version (COSMO v5.12) that is capable of using GPU accelerators by employing a rewritten dynamical core and openACC directives (Fuhrer et al., 2014; Leutwyler et al., 2016; Fuhrer et al., 2018).

COSMO solves the fully compressible atmospheric equations using finite difference methods on a structured grid (Steppeler et al., 2003; Foerstner and Doms, 2004). The model grid is a rotated latitude-longitude grid with terrain-following vertical coordinates. Time integration is performed using a split-explicit third-order Runge-Kutta discretization scheme (Wicker and Skamarock, 2002). Horizontal advection is treated with a fifth-order upwind scheme and vertical advection uses an implicit Crank-Nicolson scheme (Baldauf et al., 2011) and centered differences in space. Radiative transfer is parameterized based on a

delta-two-stream approach by Ritter and Geleyn (1992). For cloud microphysics, a single-moment scheme that parameterizes





five prognostic variables (cloud water, cloud ice, rain, snow and graupel; Seifert and Beheng, 2006) is chosen. Over land, soil processes are represented using the multi-layer soil model TERRA with eight active soil levels from 0.005 to 14.58 m (Schrodin and Heise, 2001). For the parameterization of surface fluxes, a TKE-based surface transfer scheme is used. Both the shallow- and deep-convection parameterization schemes were switched off in our model simulations. According to Vergara-Temprado

et al. (2020), switching off deep convection was found to reduce the bias in the shortwave radiative balance for simulations at resolutions higher than 25 km. The shallow convection was switched off because this reduced our too-low cloud cover bias in the south of our model domain (section 4.5).

## 2.2 Grid structures

The model grid design differs substantially between the ocean and atmosphere (Figure 1). ROMS is set up on a telescopic grid

that covers the entire Pacific basin with one pole centered over the western U.S., while the other is located over central Africa (Frischknecht et al., 2015; Desmet et al., 2022). The horizontal grid resolution ranges from 4 km off California to 60 km in the western Pacific (resulting in 604 x 518 grid points) and thus allows for a full resolution of oceanic mesoscale features off the North American west coast while still capturing large-scale teleconnections throughout the Pacific. In the vertical, the model grid contains 64 levels. The slow baroclinic time step is set to 300 s, where one baroclinic time step includes 42 fast barotropic

time steps.

The COSMO-domain encompasses the CalCS only and reaches from the Gulf of Alaska (60 °N) down to Baja California (15 °N). Over land, the domain includes coastal orography, in order to capture the drop-off effect of near-coastal winds (Renault et al., 2016b). COSMO is setup with a horizontal resolution of 7 km with 40 vertical levels and a model top of approximately 22 km, resulting in 450 x 610 grid points in the horizontal and 40 vertical levels. The model time step is set to 40 s.

## 145 2.3 Atmosphere-Ocean Coupling

ROMS and COSMO are coupled through the Ocean Atmosphere Sea Ice Soil, version 3.0 (OASIS3) coupler (Redler et al., 2010), which allows for the synchronized exchange of data fields. Coupling is performed in parallel mode and surface fields are exchanged every 600 s between the two model components. This means that the exchange takes place every second ROMS time step and every 15th COSMO time step. The transferred field thereby represents the average over the previous coupling

period. For the spatial remapping between the two very different grids, a first-order conservative remapping is applied, which implies that the weight of a source cell is proportional to the area intersected by the target cell. Land and ocean grid points are distinguished by the condition at the center of the target grid (Redler et al., 2010).

The exchange fields from the atmosphere to the ocean include the surface wind stress, the surface net heat flux, direct shortwave radiation and the total evaporation-freshwater flux. This implies that ROMS receives these fields computed on a

higher spatial resolution than from reanalysis data and at a higher temporal frequency. The ocean in turn sends to the atmosphere the ocean SST and the ocean current velocity, hence including the so-called thermodynamic and mechanical ocean-atmosphere coupling pathways (Renault et al., 2016a, 2019). While the exchange of SST from ROMS to COMSO is straightforward (the surface temperature in COSMO is overwritten by the SST from ROMS at each coupling time step), the exchange of




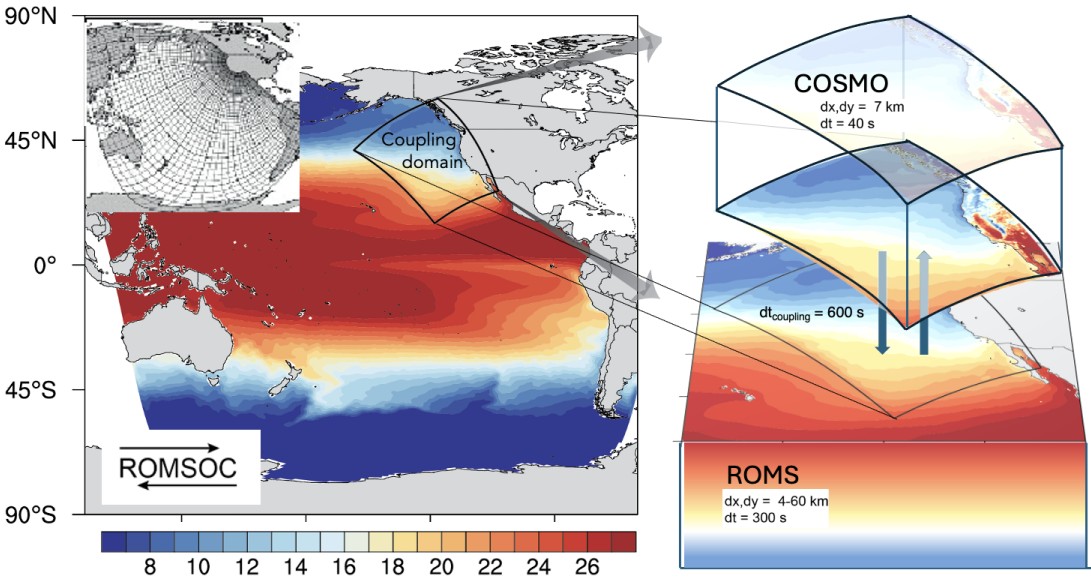

**Figure 1.** Diagram depicting the ROMSOC model setup for the northeast Pacific. While the ocean model (ROMS) covers the entire Pacific ocean, COSMO covers only the target region of the northeast Pacific. The ocean and atmosphere are fully coupled within the COSMO domain, while the ocean is running as a forced model for the rest of the Pacific. Also shown is the telescopic grid for ROMS. The colored contours show the mean SST for the upwelling season (April-August) averaged over 2010-2021.

momentum is a new feature that we added to COSMO. The ocean current velocity ($u_s/v_s$) enters the surface u and v-momentum

flux calculation in COSMO and is subtracted from the atmospheric wind ($u_a/v_a$). Hence, the inclusion of the ocean current velocity leads to a relative wind velocity ($u = u_a - u_s/v = v_a - v_s$) entering the COSMO momentum flux calculation. For the computation of the vertical momentum diffusion fluxes used to imply a zero surface velocity, we introduced an arbitrary surface boundary condition $v_s$ given by ROMS.

At the lateral interfaces of the coupling domain, a transition layer of 7 COSMO grid points is included in ROMS, to allow for

a smooth transition from forced to coupled mode. Within this transition layer, we apply a weighting coefficient to the exchange fields.

## 2.4   Numerical experiments

Here we present results from two model experiments, one from our fully coupled model setup (including thermodynamical and mechanical coupling) and one from our ocean-only setup for comparison. These hindcast simulations cover a 12-year period

and start on January 1st of 2010 extend through December 31st of 2021.

The initial conditions for COSMO for the year 2010 are taken from ERA5. In contrast, ROMS starts from a restart file of an ocean-only simulation that was run for 31 years prior to the model start (Desmet et al., 2022).



For the atmospheric forcings at their open lateral and surface boundaries, both the ocean and atmosphere models use ERA5 reanalysis data (Hersbach et al., 2020). COSMO uses at its lateral boundaries ERA-derived atmospheric and surface temperature, 3D wind fields, specific humidity, cloud liquid water and ice content, pressure, atmospheric ozone and aerosol content, root depth, vegetation fraction, the leaf area index and the snow temperature and thickness at 6-hour frequency. ROMS is forced daily at the surface outside the COSMO region by surface short- and longwave radiation fluxes, wind stress and surface freshwater fluxes. In addition, ROMS SST and sea surface salinity (SSS) fields are being restored to monthly Reynolds SST fields (NOAA OI v2) and climatological SSS (ICOADS, Worley et al., 2005). This restoring is turned off in the coupled domain. The oceanic forcings for ROMS at the open lateral boundaries in the Southern Ocean are based on monthly data (Frischknecht et al., 2015). Potential temperature and salinity are taken from the World Ocean Atlas 2013 (Levitus et al., 2014), while currents and sea surface height (SSH) stem from SODA 1.4.2 (Carton and Giese, 2008).

## 3 Compute setup and Performance

In order to make efficient use of the hybrid CPU-GPU architecture of the Piz Daint supercomputer, ROMSOC uses the full node capacity comprising 12 CPU (Intel Xeon E5-2690 v3 @ 2.60GHz) and 1 GPU (NVIDIA Tesla P100). While the ROMS model runs entirely on CPU, COSMO has been fully ported to GPU except for the I/O and coupling parts (Figure 2a). To achieve this, the dynamical core has been rewritten using GridTools, a GPU enabling software library that was developed specifically for weather and climate models (Afanasyev et al., 2021). The remaining parts of the model, mainly 1D physics, are handled using OpenACC compiler directives. For a detailed description of the GPU-enabled COSMO version, we refer to Fuhrer et al. (2014, 2018) and Leutwyler et al. (2016).

Figure 2b shows the strong scaling results for ROMSOC on setups ranging from 4 to 84 nodes, with 4 nodes being the lowest node number required for ROMSOC to run. The coupled model scales very well up to 64 nodes, where it reaches a parallel efficiency of 80.8%. Beyond 64 nodes, the additional gain in speed-up is small and the parallel efficiency is reduced to 77%. Therefore, we consider a setup with 64 nodes as ideal. The most relevant model processes (pure runtime, initialization, and output writing) as well as the individual runtimes for ROMS and COSMO and a ROMSOC CPU setup for the same number of nodes are presented in Table A1. As all benchmarking simulations were performed on the Lustre /scratch system of Piz Daint, which entails some degree of system variability by design, we report all specifics for the ideal model setup as a mean of three test simulations including the standard deviation. As evident from Table A1, there is a significant improvement in model runtime due to the joint CPU/GPU system. Model initialization and output writing take up a substantial fraction of the total runtime, where the latter is however highly dependent on the number of output variables. Model initialization time decreases in relative terms for longer simulations, such as our hindcast.

In their uncoupled setup (but on the same node configuration), COSMO is twice as fast as ROMS (Table A1), indicating that despite the 7.5-larger time stepping interval in ROMS, ROMSOC is constrained by the oceanic part of the model running on CPU only. Thus, adding the atmospheric component to the coupled model only marginally affects the overall runtime on a heterogeneous system architecture, as for ROMSOC both CPU and GPU of each node are used without any processors idling.



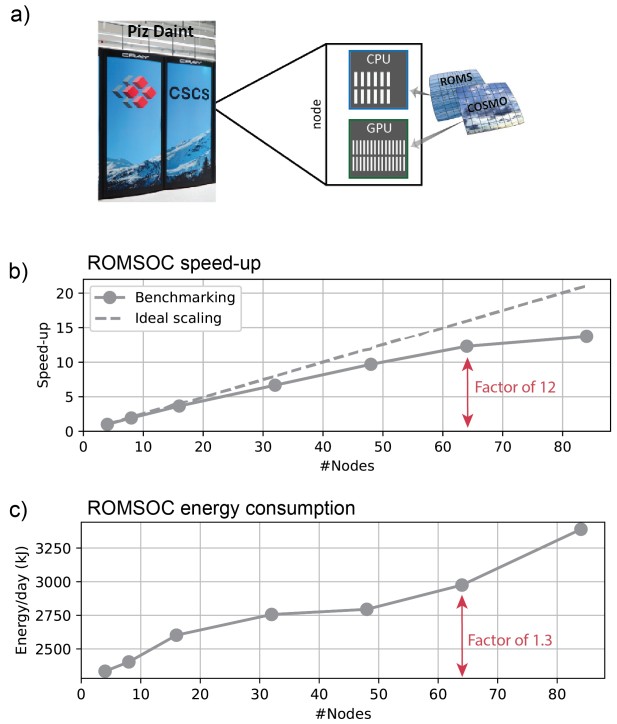

**Figure 2.** a) ROMSOC architecture on Piz Daint. b) ROMSOC benchmarks after a simulated period of 4 days, without output writing. c) ROMSOC energy consumption per simulated day for the simulations depicted in b).

Note though that ROMS is coupled to BEC, which substantially increases its runtime. This slowdown of model performance on CPUs is also shown in the runtime when ROMSOC is run on CPUs only: the model runtime increases by a factor of 6 as compared to the same setup on 64 nodes running on GPU.

210    In addition to the gain in runtime, the energy consumption is also greatly reduced when using GPUs (as evident from the resource utilization report). The 4 day COSMO simulation ([5] in Table A1) used only one third of the energy used by the 4 day ROMS simulation ([6] in Table A1) and the energy consumed by the CPU-GPU ROMSOC simulation ([1] in Table A1) is five times smaller than that consumed by the CPU-only simulation ([7] in Table A1). This efficient scaling of energy usage is also shown in Figure 2c, where the increase in energy consumption for the 64 node setup increases by only a factor of 1.3 215    with respect to the simulation on 4 nodes (as a result of more nodes being used), while the speed-up in runtime increases by a factor of 12 (Figure 2b). Such an increase in energy efficiency lowers the already high energy footprint of climate simulations and hence their attributed costs.

Given the scientific benefits from adding an interactive atmosphere to the ocean model (see Section 4), running a coupled model over an ocean-only should be taken into consideration if a heterogeneous CPU-GPU computing system is available.





## 4 Model evaluation

In the following, we evaluate the fidelity of our 12-year coupled hindcast simulation against a number of observational constraints. To assess the impact of using the coupled model, we contrast the results with those from an uncoupled ROMS simulation forced with daily ERA5 reanalysis. We focus on the region between 35°N and 47°N and distinguish between a 100 km wide coastal band, and the offshore region (see Figures 3a and A1 for regional boundaries).

### 4.1 Sea-surface temperature

Figure 3 shows the strengths but also the limits of ROMSOC in simulating the SST observed in the Northeast Pacific by Reynolds (Reynolds et al., 2007) and ERA5. Overall, the coupled model succeeds in representing the large-scale SST gradients (Figure 3a-c), especially the strong onshore-offshore gradient between the US west coast and the open ocean, primarily driven by coastal upwelling. The maps in Figure 3a-c, which actually show the climatological SST pattern associated with the upwelling season (April-August), clearly reveal the band of relatively cold SSTs around 12°C along the Californian coast, with fine-scale structures in the models not visible in the coarser resolution (0.25°) observational products.

However, ROMSOC has a cold SST bias of about 1-3°C compared to the observations during this time period (Figure 3d,e). This cold bias persists throughout the year, but is substantially smaller (1°C) outside of the upwelling season. There is also a persistent cold SST bias in the offshore region of about 2°C (Figure 3e), with the fall period having the largest offsets. An additional negative SST bias occurs in the northern part of the domain close to the COSMO model domain edge.

The biases simulated by the coupled model have a relatively limited impact on the model's ability to simulate interannual variability (Figure 4), both in the coastal and in the offshore regions. ROMSOC simulates well the substantial year to year variations in SST, especially the 2013-2015 Northeast Pacific "blob" heatwave event when the SSTs in the coastal regions of the CalCS were more than 2°C warmer than usual. Also the SST variations in the offshore regions are well captured (Figure 4b). The SST biases in the coastal region (Figure 4a,c) peak during the upwelling season, as already seen in the climatological analysis (Figure 3), but these biases do not vary substantially with the interannual anomalies. There is a slight tendency in the coastal region for the biases to get smaller during abormously warm years. This is particularly evident during the "blob" heatwave event, where the biases nearly disappeared (Figure 4a,c). In contrast, the biases stay nearly constant in the offshore region (Figure 4b,d). Also noteworthy is that there is no long-term drift in our model simulation.

Figures 3 and 4 also reveal that the SST biases in the coupled simulation are larger than those in the uncoupled simulation, suggesting a loss of fidelity when switching from the uncoupled to the coupled simulation. An important reason for this increasing bias is our turning off of the SST restoring in the coupled simulation. In the uncoupled simulation, this Newtonian restoring keeps the model's SST closer to the observation, while in the coupled simulation, the model's SST is permitted to evolve freely. This effect is particularly strong in the offshore region, as the Newtonian restoring there is very successful in bringing the model simulated SST close to the observed one.



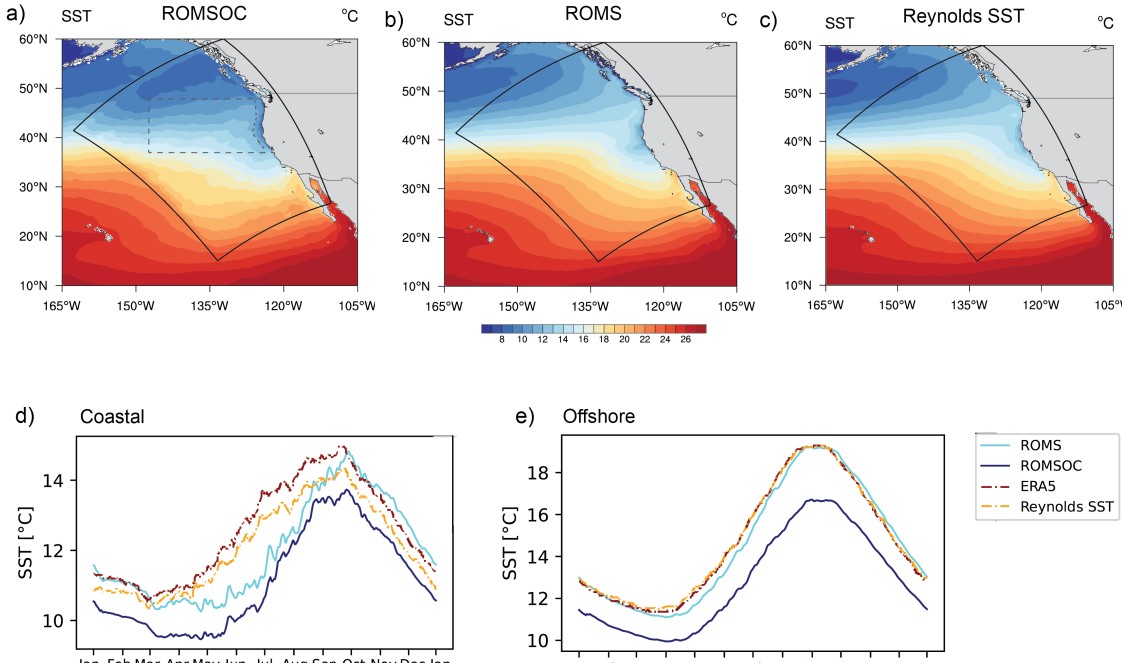

**Figure 3.** Assessment of the model simulated climatology of sea-surface temperature (SST): a) Maps of the ROMSOC simulated SST averaged over the upwelling season, that is from April to August b) as a), but for ROMS only and c) as a) but for the observation-based Reynolds SST product (Reynolds et al., 2007). The data shown are climatologically averaged over all 12 simulated years, for 2010 through 2021. Also shown in panel a are the coastal and offshore analysis region in the California Current System. d) Timeseries showing the annual cycle of simulated and observed SST averaged over a 100-km band along the US coast (shown in panel a). e) As (d), but for the offshore region shown in panel a.

## 4.2 Winds

ROMSOC simulates the observed large-scale structure of the 10-m winds well, especially the contrast between the strong, upwelling-favorable wind regime along the west coast, and the weak winds in the offshore region (Figure 5). Compared to the 10-m winds in the coarser resolution ERA5 product, ROMSOC includes more details, especially nearshore, where wind speeds are around 2 m s$^{-1}$ stronger as for the observational products. The 10-m wind speed is also increased in the north of the domain, possibly favoring advection of cold water masses, which could lead to overall colder SST in the north of the coupling domain in ROMSOC (Figure 3).

Focusing on one particular strong-wind event on March 7th, 2011, Figure 6a,b highlights the increased level of detail in representing the coastal winds in ROMSOC as compared to ERA5. As ERA5 is used as atmospheric forcing for ROMS, this





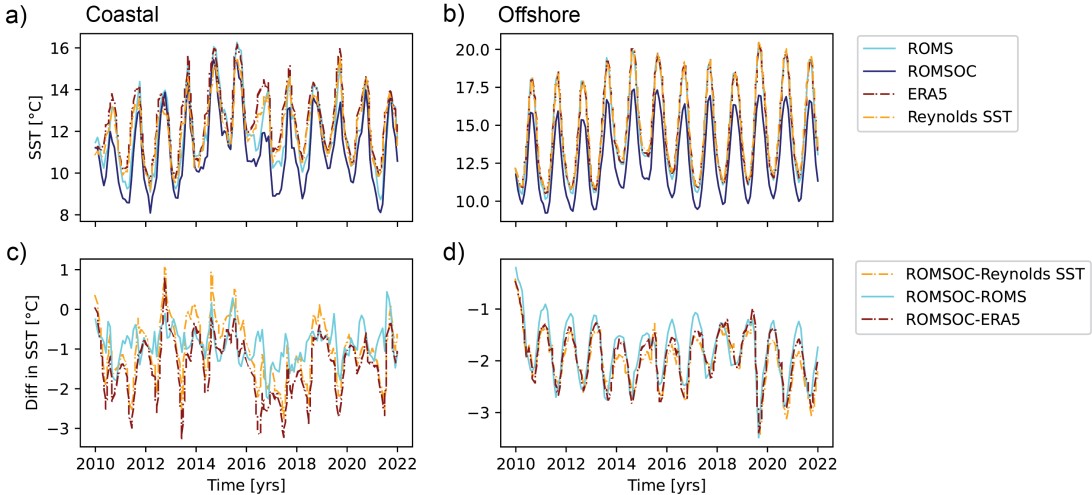

**Figure 4.** Evaluation of interannual variability: a) Comparison of model simulated and observed SST over the hindcast period averaged over the coastal region (see 3a). b) As a) but for the offshore region. c) Difference between simulated and observed SST for the coastal region. d) As c), but for the offshore region.

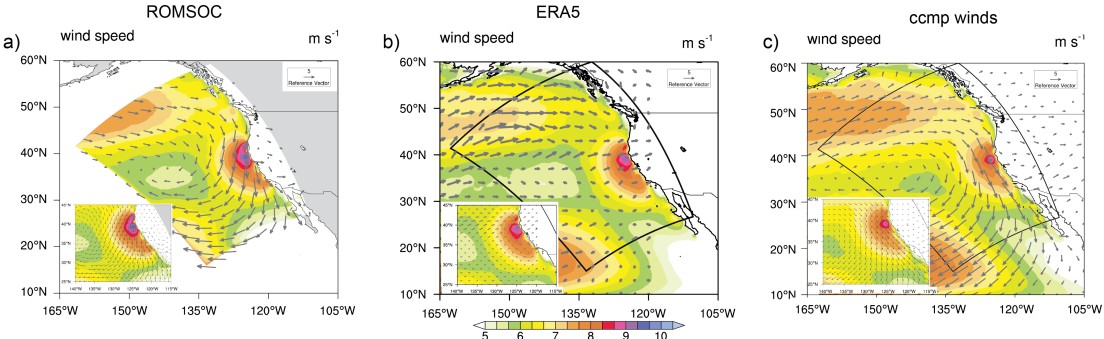

**Figure 5.** Evaluation of 10-m windspeeds over the upwelling season. a) Map of the windspeed as simulated by ROMSOC. The inlet shows a detailed map for the coastal region. b) As a) but for the coarser resolution reanalysis produce ERA5 (Hersbach et al., 2020). c) As a) but the for the satellite based CCMPv2 data (Atlas et al., 2011; Mears et al., 2019). Shown are the averages over all years (2010-2021).

increased level of details is not passed forward to the ocean model. Resulting from the higher resolution of ROMSOC, smaller-scale regions with increased windspeed are visible along the North American coast as compared to ERA5. Such events are essential for triggering coastal upwelling and upper ocean mixing.

Figure 6c shows the coastal wind drop-off, which describes the slackening of the alongshore winds close to the coast and is essential for coastal upwelling (Renault et al., 2016b). This drop-off is represented in both ROMSOC and ERA5, however, the



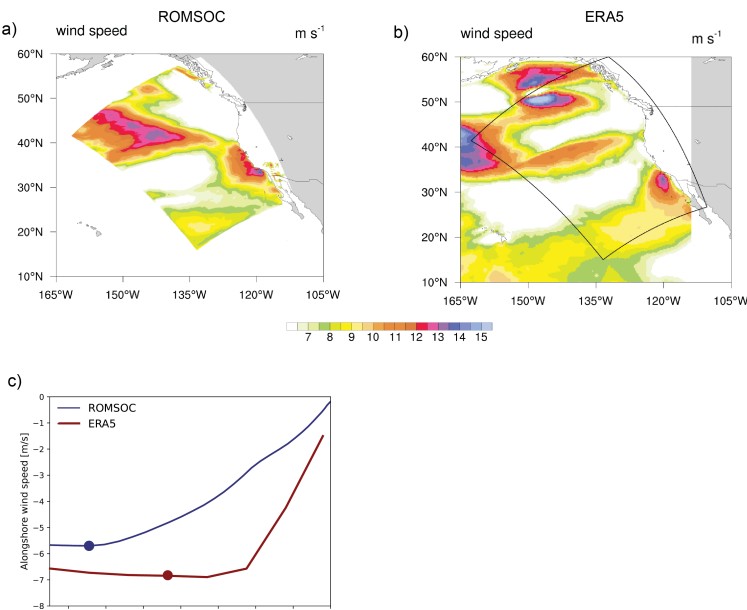

**Figure 6.** Snapshot of 10-m windspeed on March 7th 2011 in a) ROMSOC and b) ERA5. c) Wind drop-off averaged over the upwelling season from 2010-2021 in ROMSOC and ERA5 for a band between 40-42°N. The dots show the local minimum of the alongshore wind, the so-called drop-off length (129 km for ROMSOC and 86 km for ERA5).

drop-off from 50 km onwards is more gradual in ROMSOC and more sharp in ERA5, as this distance represents around only 2 ERA5 grid boxes.

Similarly to the 10-m wind speed, ROMSOC simulates a more positive wind stress curl (Figure 7), enhancing the coastal upwelling further. This positive bias in wind stress curl could arise from the generally stronger winds, the interaction of the wind with the topography and the representation of lateral wind speed gradients. In the coarser-resolution ERA5 and CCMPv2 products, these interactions may be less well-represented and the coastal wind speeds are generally lower.

At the same time, there are also some shortcomings. In particular, the strong coastal 10-m wind in ROMSOC (Figure 5a) relative to ERA5 and CCMPv2 produces too much coastal upwelling of cold waters from below, contributing strongly to the cold SST bias as compared to ROMS in the coastal region. ROMSOC simulates these stronger winds as a consequence of the higher spatial and temporal resolution of COSMO compared to ERA5. This results in a better resolution of small-scale variations such as the wind's interaction with the adjacent topography and short-lasting peaks in wind speed from passing storms as well as the representation of the wind drop-off of alongshore winds. On the other hand, the parameterization of surface roughness, cloud cover, and hence lower boundary layer turbulence can influence the wind speed in ROMSOC and contribute to differences between the model and the observational products.





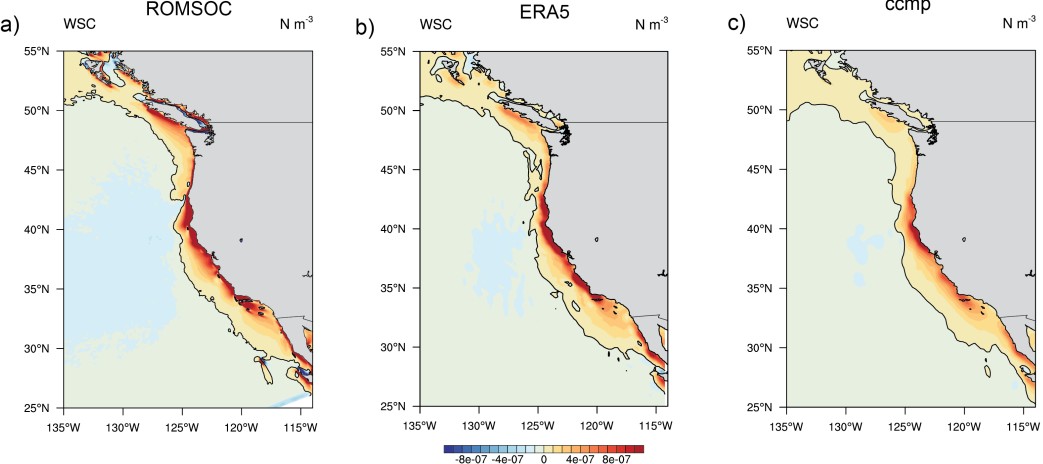

**Figure 7.** Evaluation of the 10-m windstress curl averaged over the upwelling season. a) Map of the ROMSOC simulated wind stress curl. b) as a) but for ERA5. c) as a) but CCMPv2 data. The zero-windstress curl contour is added in black. Shown are the averages over all years (2010 - 2021).

## 4.3 Mixed-layer depth

The distribution of mixed layer depths (MLD) are notoriously difficult for models to simulate correctly. In the northeast Pacific (Figure 8), the annual cycle of MLD is characterized by a well-defined MLD shoaling in summer when waters are warmer and more stratified, and MLD deepening in winter, as passing storms and the resulting higher wind stress mix the upper ocean (Jeronimo and Gomez-Valdes, 2010) (Figure 8a,b). This annual cycle is reproduced well by both ROMSOC and ROMS, with an up to 20 m deeper MLD in ROMSOC and a more pronounced annual cycle (note that we diagnose MLD from ROMS

KPP-scheme (see Section 2.1), i.e., it reflects an active mixing layer). The deeper MLD in ROMSOC can be seen close to the coast as well as throughout the domain, where it agrees better with Argo float observations (Figure 8b). Here, ROMSOC actually improves the simulations relative to ROMS.

However, even though the SST pattern is extremely smooth along the COSMO domain edge, we may see influences from the interpolation between COSMO and ERA5 along the domain edges and thus any signals close to the boundaries may be

artificial rather than physical. Nevertheless, Argo floats observe a deeper MLD throughout the domain and hence estimates from our coupled model are closer to observations than those from ROMS, indicating again the dominant effect of wind-driven mixing and the high importance of high resolution wind forcing.

## 4.4 Ocean circulation

Ocean mixing does not only occur in the vertical, but is also essential in the horizontal direction through mesoscale ocean

eddies or larger-scale ocean currents. Off California, the large-scale flow is characterized by the cold-water California current, which moves southward along the Californian coast. Right along the continental edge, the narrower California Undercurrent



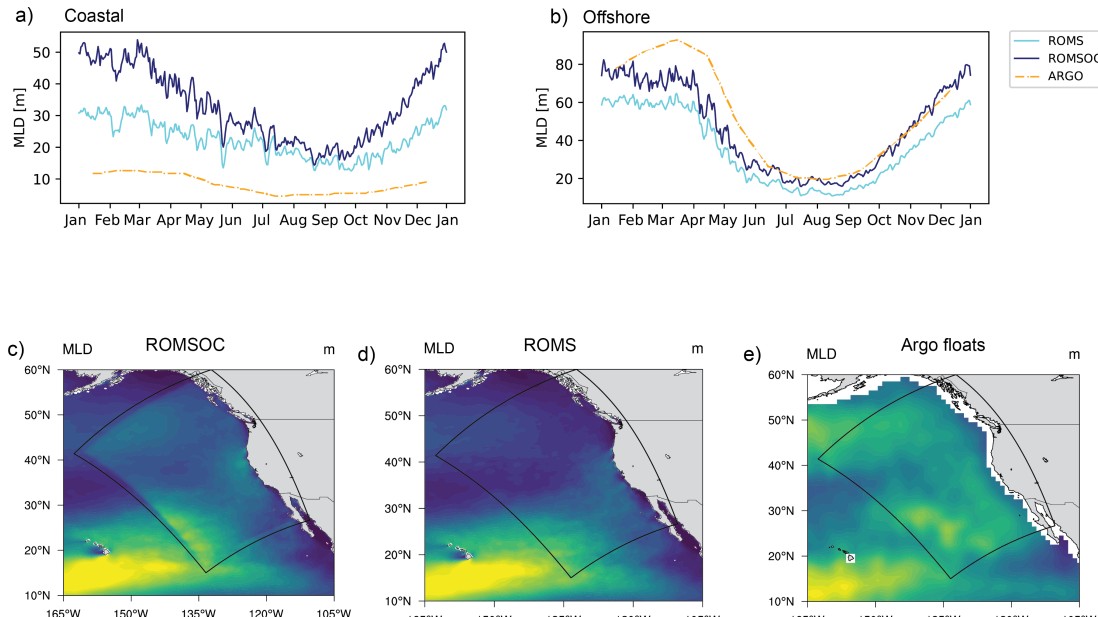

**Figure 8.** Assessment of the MLD. a) Seasonal cycle of the model simulated and observed MLD for the coastal region, defined as a 100-km band along the US coast. b) as a) but for the offshore region. c) Map of ROMSOC simulated spatial patterns of MLD averaged over the upwelling season of all years. d) as c) but for ROMS. e) as c), but based on product derived from Argo float observations (Wong et al., 2020). Note that due to the coarse resolution of Argo close to the coast, the quality of Argo data in this region is reduced.

transports warmer water northward. Further offshore, mesoscale ocean eddies spin off the California current and transport coastal waters into the open ocean (Kurian et al., 2011).

Due to the high (4 km) horizontal resolution of the ocean model close to the coast, our model allows these features to develop
close to the coast and propagate westward, as seen in snapshots of daily sea surface height (SSH, Figure 9a,c). Ocean eddies are visible in both ROMSOC and ROMS, with both anticyclonic and cyclonic eddies occurring at similar densities in both model simulations (Figure 9d). The EKE averaged in a band off California (as outlined in Figure 9a) is lower in ROMSOC as compared to ROMS, although not statistically significant. Yet, we suspect the coupling of momentum in ROMSOC to be responsible for this effect, which has been found to reduce ocean EKE compared to uncoupled simulations and lead to a more
realistic simulation of EKE in coupled simulations (Renault et al., 2016c).

Apart from ocean eddies, the coupling to the wind also impacts ocean currents. Figure 10 shows vertical transects of the geostrophic alongshore velocity at 36°N in ROMSOC and ROMS and their difference, respectively. Here, we excluded a comparison to observations, as within the relatively short averaging period of 12 years the high interannual variability through passing eddies becomes too dominant, which leads to large discrepancies between our models and the observations (Figure
B1), which was not emerging in the longer averaging period applied by Frischknecht et al. (2018). Both the coupled and uncoupled model represent the location and strength of the California Undercurrent, located right at the continental edge, and





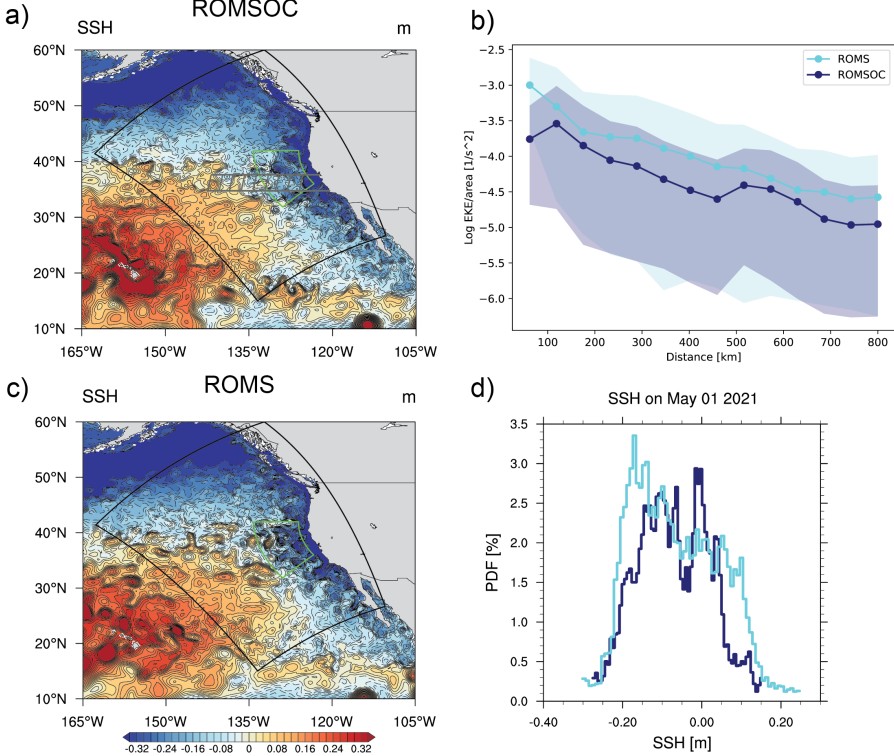

**Figure 9.** Snapshot of sea surface height on May 1st 2021 in a) ROMSOC and c) ROMS. b) Logarithmic eddy kinetic energy as a function of distance from coast averaged over the upwelling season for 2010-2021 in ROMSOC and ROMS. The values are normalized by area. The shading represents the standard deviation over the 12-year hindcast. d) PDF of a snapshot of SSH from the same day as shown in a) and c) in ROMSOC and ROMS. The grey and green areas outlined in a) denote the area of averaging for b) and d), respectively.

the relatively slower and more surface-confined California Current. The California Undercurrent is stronger (up to 0.04 m/s) in ROMSOC as compared to ROMS, which further increases the bias representing the Undercurrent in ROMS identified by Frischknecht et al. (2018). The California Current is also slightly stronger in ROMSOC as compared to ROMSOC by up to 315 0.02 m s$^{-1}$, which is in turn reducing the too-weak bias in the California Current strength in ROMS (Frischknecht et al., 2018).

## 4.5 Cloud patterns and radiative fluxes

The previously discussed oceanic properties resemble the atmospheric mean patterns and variability as simulated by COSMO and ERA5 as forcing for ROMS. In this regard, Figure 11 shows the total cloud fraction in ROMSOC, ERA5 and MODIS Terra satellite data. Also outlined is the fraction of low clouds in Figure 11a.

In ROMSOC, the high cloud fraction in the north of the domain with the largest fraction of low clouds is well-represented. Cloud fraction decreases towards the south, reaching 0.2 off southern California. While this gradient in cloud cover is also seen



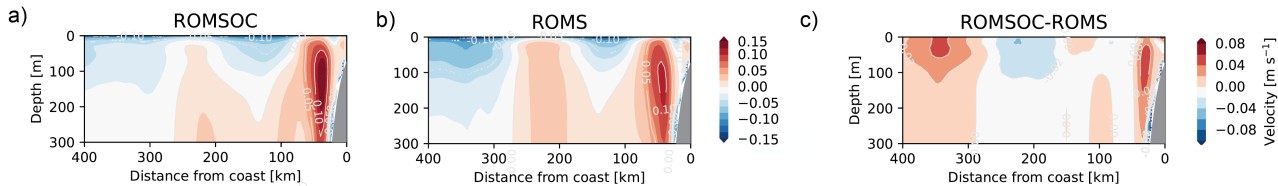

**Figure 10.** Vertical transect of geostrophic alongshore velocity averaged over the upwelling season in a) ROMSOC, b) ROMS, and c) ROMSOC-ROMS along the 66.0 Line of the CalCOFFI observations (corresponding to 121.8-125.7°W/36°N). Positive (negative) velocities denote northward (southward) flow.

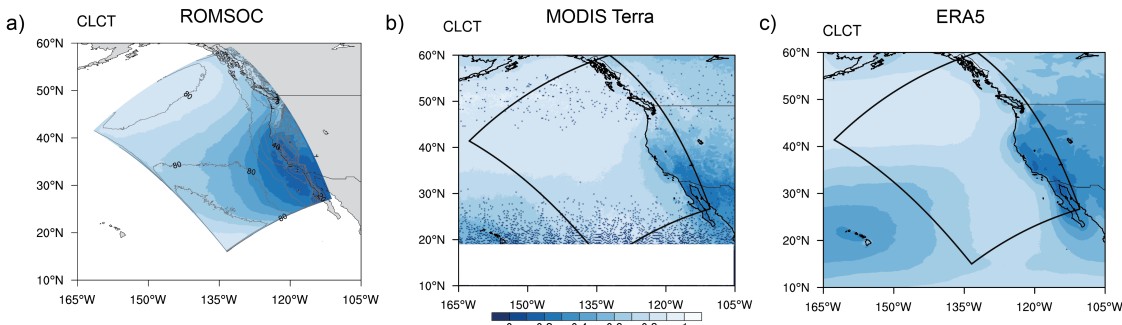

**Figure 11.** Total cloudfraction averaged over the upwelling season of all years in a) ROMSOC and b) Modis terra satellite data, and c) ERA5. The contours in a) display the fraction of low clouds of the total cloud amount shown in colors.

in the observations, the area with low cloud cover stretches out further offshore in ROMSOC, leading to a bias with too low cloud cover compared to satellite data and also ERA5. However, this bias in cloud cover does not lead to enhanced incoming shortwave (SW) radiation at the surface in that region as compared to ERA5 (Figure 12a,e) and reflects the very common
too-few-too-bright cloud bias known from global models (e.g., Nam et al., 2012). This lower SW radiation in the south and especially southwest in the COSMO domain contributes to driving the colder offshore SST in ROMSOC, as less SW radiation is available to heat the upper ocean.

Turbulent surface fluxes exhibit similar patterns in ROMSOC and ERA5, with a sensible heat flux from the atmosphere into the ocean and a reversed weak latent heat fluxes in the north of the domain (Figure 12c,d,g,h). Here, the overlying warmer
atmosphere heats the relatively cold upper ocean through sensible heat fluxes. Similarly, along the California coast in the upwelling band, sensible heat fluxes are directed towards the colder ocean. In the south of the domain, the sensible heat flux switches sign and the relatively warmer ocean looses heat to the overlying atmosphere. This effect is stronger in ROMSOC,





further contributing to the relatively colder offshore SST in ROMSOC. The latent heat flux generally agrees well between ROMSOC and ERA5, with slightly too weak latent heating in the south of the domain in ROMSOC. This weaker latent heat

flux in ROMSOC could be due to too little evaporation from the colder SST or the lack of stratocumulus clouds in that region, which generally tend to dry our the marine boundary later through entrainment of dry free tropospheric air to lower levels and hence increase the latent heat flux (Stevens, 2007).

## 5    Discussion

Overall, our coupled model provides a realistic view of the oceanic and atmospheric dynamics off California. The large-scale

SST structure and coastal upwelling are overall well-represented in ROMSOC, however, with a cold bias of 1-3 °C compared to the observational products. We related this cold bias to stronger wind forcing in the coupled model as compared to its uncoupled counterpart. At the same time though, this strong wind forcing leads to increased ocean mixing and a deeper and more realistically simulated MLD throughout the domain in ROMSOC as in ROMS compared to Argo observations.

The coastal cold bias in ROMSOC is comparable to the cold bias in the regional coupled simulations by Renault et al.

(2020), which the authors also relate to the coarser resolution and biases in the observational product due to coastal cloud cover. Simulating the correct strength and extent of the coastal upwelling has been shown to be generally difficult (Renault et al., 2020, and references therein), mainly due to uncertainties in the atmospheric forcing (surface stress, radiative fluxes and cloud cover). As these processes remain parameterized in regional models with resolutions on the order of kilometers, reducing such biases on the small scale remains to be challenging. With regard to global models, the upwelling is well-represented and

ROMSOC does not reproduce the too-warm SST bias as shown in Richter (2015).

However, this cold SST bias is not seen in the uncoupled ROMS simulation, as a result of the applied SST restoring in ROMS. While the SST restoring is a useful approach for constraining modeled SST (i.e. preventing model drift, improving model accuracy, compensating for limitations in the atmospheric forcing data, and enhancing the skill of physical oceanic parameterizations depending on SST), its application has limitations for some scientific applications of regional models. As

an example, we refer to future model projections using regional models, where the model is forced with perturbed boundary conditions to simulate a future climate on a higher resolution as possible using global models. As obviously no observations for a future climate are available, SST restoring would not be possible (except restoring to modeled SST from global models, which, however, have significant biases on the smaller scale). Such a study using ROMSOC is currently in preparation. Hence note that by comparing ROMSOC to its uncoupled counterpart ROMS for two hindcast simulations, we are not comparing the

models under exactly the same premises and the advantages and disadvantages of the applied SST restoring in ROMS should be kept in mind.

For the improved modeled processes in ROMSOC, such as the MLD representation and partly the ocean circulation, we want to highlight that both the spatial and temporal resolutions of COSMO are higher compared to ERA5 and that the coupling time step of ROMSOC is 144 times higher than the forcing time step applied to ROMS. Hence, we cannot identify with certainty

whether the spatial or temporal resolution of the wind forcing is responsible for oceanic differences between ROMS and



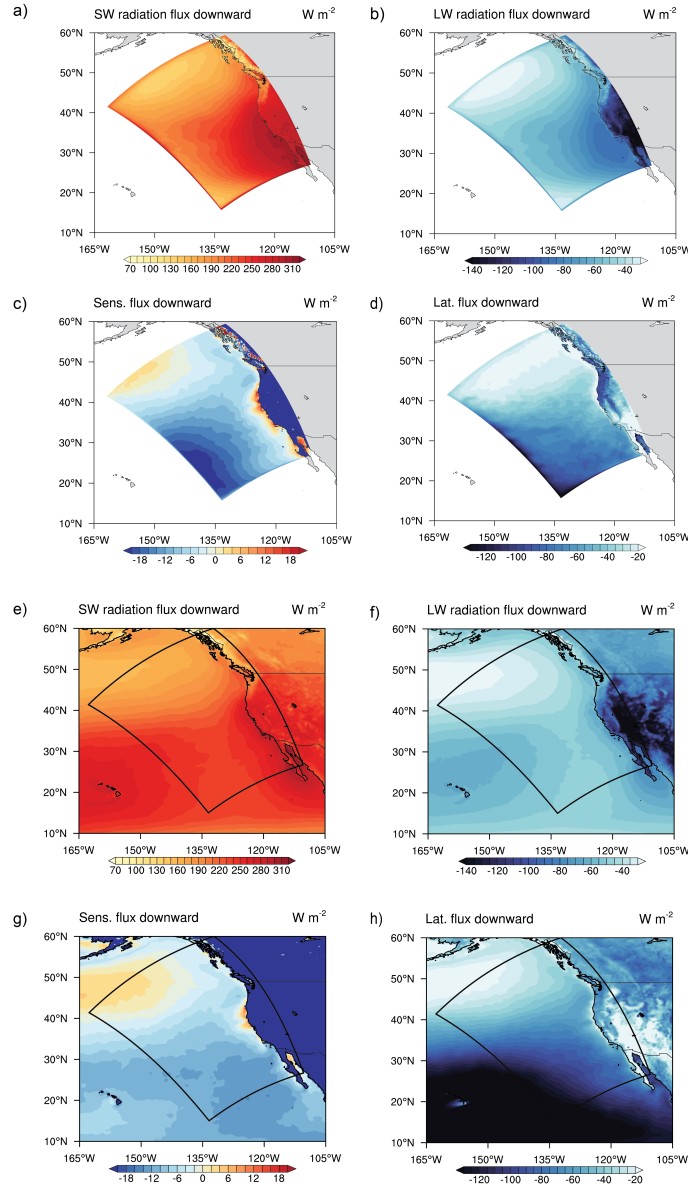

**Figure 12.** Radiative fluxes averaged over the upwelling season of all years with a,e) downwelling shortwave radiation, b,f) downwelling longwave radiation, c,g) downwelling sensible heat flux and d,h) downwelling latent heat flux for ROMSOC and ERA5, respectively.

ROMSOC. However, we performed a sensitivity simulation for one year where we applied the coupling only daily in ROMSOC. The difference in MLD between our default ROMSOC simulation and the sensitivity run is mainly positive, indicating a 8-10 m deeper MLD in ROMSOC with a higher temporal coupling resolution (Figure C1), which represents about the average





difference between ROMSOC and ROMS during the upwelling season (Figure 8a). Hence, we suggest that the temporal
resolution of the coupling is mainly responsible for forcing stronger ocean mixing and coastal upwelling in the coupled model.

In addition to the importance of surface forcing for SST and MLD, lower EKE is simulated by ROMSOC as compared to
ROMS for the CalCS (Figure 9b). As first pointed out by Renault et al. (2016c), momentum coupling, i.e. the passing of the
ocean current velocity to the surface u and v-momentum flux calculation in coupled models is responsible for this more realistic
representation of oceanic mesoscale eddies. This effect acts on the surface stress and has been shown to have a counteracting
effect on the wind itself (Renault et al., 2016c), thereby acting as an oceanic eddy killer, reducing the surface EKE by half. The
reduction of surface EKE in our coupled simulation is smaller, but nevertheless visible. More detailed analyses to quantify the
effect of momentum coupling are needed in the context of future studies for our coupled model, but the reduction in EKE in
ROMSOC already represents one of the advantages of using coupled over uncoupled models when investigating upper ocean
mixing.

Regarding the atmospheric conditions, our coupled model features a bias in low cloud cover, which is very common among
models. This bias is in turn affecting the surface fluxes. On contrary, Renault et al. (2020) simulated too high cloud cover
off California, highlighting the strong sensitivity of cloud cover to parameterizations and tuning parameters. We suggest that
higher vertical and horizontal resolutions of the atmospheric model and/or more sophisticated cloud parameterizations (e.g. a
two-moment cloud microphysics scheme) could help reduce this bias, but were outside the scope of this work.

**6 Conclusions**

Here we presented the model specifics, setup, performance, and a model evaluation of our newly developed coupled atmosphere-
ocean model (ROMSOC), capable of running on a hybrid system architecture using both CPUs and GPUs. By running the ocean
model ROMS on CPU and COSMO on GPU, we can make efficient use of this type of system architecture with a gain in per-
formance by a factor of 6 for the same number of nodes (as COSMO uses the available GPU that would otherwise be idling).
Similarly, we can conclude from the similar runtimes for the model in coupled- and uncoupled mode that the additional cost
by running in coupled mode is small, as the compute nodes are fully used by the coupled model setup.

Results from our hindcast show that coastal as well as offshore SST are colder in the coupled model compared to a ROMS-
only simulation. This results from stronger upwelling along the coast of California, a stronger wind-driven mixing, and the
applied SST restoring in ROMS, constraining SST for the uncoupled model. For simulating upper ocean mixing, the coupled
model simulates a more realistical MLD. Here, ROMSOC benefits from the higher temporal resolution of the coupling as
compared to the ERA5 forcing driving ROMS, which better resolves the surface wind stress forcing. In addition, our newly
implemented coupling of ocean momentum in ROMSOC, leads to a small reduction in eddy kinetic energy, and potentially a
more realistic representation of oceanic mesoscale eddies according to previous studies. Our coupled model simulates cloud
cover in the north of the domain very well, with high cloud cover throughout the upwelling season. In the south of the domain
our model shows a lack of cloud cover, a very prominent bias in climate models. This shortcoming is however compensated by
too bright clouds and hence no large bias in incoming SW radiation.





Our coupled model is ready to be used for investigating a range of scientific questions comprising both the ocean and atmospheric realm. Currently, analyses regarding the atmospheric influence on marine extreme events are underway. As these analyses focus mainly on the north part of the domain, the cloud bias in the south as well as the cold SST bias in the upwelling

area becomes less significant. In addition, our model is set up to perform future simulations for the CalCS following the pseudo global warming (PGW) approach as presented by Brogli et al. (2023), which will be presented in a follow-up study.

*Data availability.* Processed data and scripts are available here: https://doi.org/10.3929/ethz-b-000693509. Raw model output is archived and will be made available on request.

*Code and data availability.* The ROMS model code is available through: https://github.com/UCLA-ROMS/Code. The particular version of

the COSMO model used in this study is based on the official version 5.12 with many additions to enable GPU capability. We will make this code version available on request for the reproducible of this study.



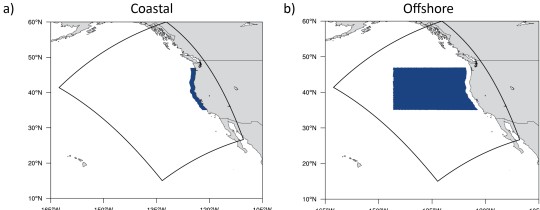

**Figure A1.** Masks used for averaging for the a) coastal (100-km band along the coast) and b) offshore region.

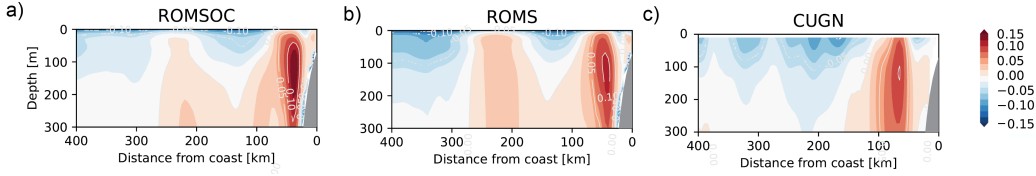

**Figure B1.** Vertical transect of geostrophic alongshore velocity averaged over the upwelling season in a) ROMSOC, b) ROMS, and c) CUGN observations along the 66.0 CalCOFFI Line (Rudnick et al., 2017).

## Appendix A: Additional Figures

*Author contributions.* GKE performed the simulations, the analyses and created the initial draft of the manuscript. ML implemented the coupling routines in the COSMO and ROMS model codes. GKE, ML and MM contributed to further develop these routines. NG acquired
the funding for this research and conceptualized the initial idea for this work. All authors were involved in developing the methodology for this study and contributed to the writing of the final manuscript.

*Competing interests.* The authors declare that there are are no competing interests.

*Acknowledgements.* This work was supported by a grant from the Swiss National Supercomputing Centre (CSCS) under projects s1057 and s1244. Funding for the scientific work came, in part, from the Swiss National Science Foundation under Grant/Award Number: 175787.
We thank also the Center for Climate Systems Modeling (C2SM) for its support in the development of the coupled model. In addition, the authors would like to thank Christoph Heim for many useful discussions and insights on the COSMO model setup, Damian Loher and Flora Desmet for help with the ROMS input data, and Helena Kuehnle for contributing to the analyses. GKE further acknowledges William T. Ball for his scientific input and his valuable contributions and ideas throughout the past years, which make him live on in our minds and hearts.



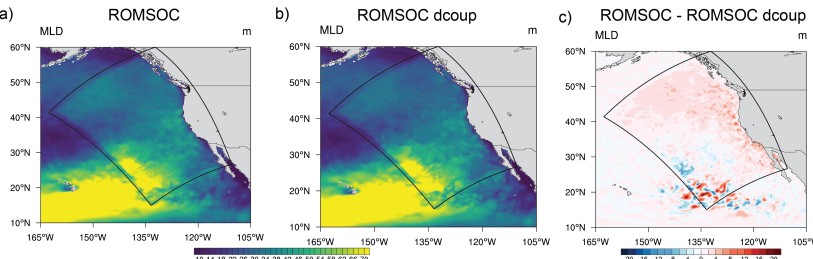

**Figure C1.** Spatial patterns of MLD averaged over the upwelling season of 2010 in a) ROMSOC, b) ROMSOC with daily coupling (ROMSOC dcoup), and c) the difference between the two model simulations.

**Table A1.** Runtime specifics for the ideal setup (64 nodes) for a 4 day simulation. The numbers in round brackets indicate the calculation, and the numbers in squared brackets indicate the numbering of the specific process. Due to the inherent variability of the file system, for process [1], [2], [4], [5], [6] and [7] the simulation has been run three times and the numbers represent the mean as well as the standard deviation across the different simulations. Note that also for COSMO-only, ROMS-only and ROMSOC on only CPU the simulations have been performed on 64 nodes.

| Process | Timing [s] |
| --- | --- |
| model initialization and runtime [1] | 858.9±2.0 |
| model initialization and runtime for 2 days [2] | 507.3±48.2 |
| mean runtime (1-2) [3] | 351.6 |
| mean initialization (2-3) | 155.7 |
| model initialization, runtime and output writing (1 + output) [4] | 1460.9±32.6 |
| mean output writing (4-1) | 602.0 |
| COSMO initialization and runtime [5] | 380.8±12.2 |
| ROMS initialization and runtime [6] | 776.2±23.7 |
| ROMSOC (CPU) model initialization and runtime [7] | 5145.7±35.0 |

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
