# Peer review of "ROMSOC: A regional atmosphere-ocean coupled model for CPU-GPU hybrid system architectures"

_EGUsphere, 2024_

## Referee Comment (RC1)

Review of "ROMSOC: A regional atmosphere-ocean coupled model for
CPU-GPU hybrid system architectures" by Eirund, Leclair, Muennich, and Gruber.

General comments

This work presents the regional coupled atmosphere ocean land model ROMSOC, which is based on the existing components COSMO for the regional atmosphere and land, ROMS for the regional ocean, and OASIS3 for the coupling. While similar such models have been constructed earlier, this article points out a novel practical implementation, that is the hybrid computational setup with the ocean component running on CPUs and the atmosphere component running on GPUs of the same nodes on the computing system they have available. In their setup, where the ocean model is slower, than the atmosphere, and the billed computing costs depend only on the number of nodes and length of computation, this means that the computations for the atmosphere and land are essentially for free. Thus the article shows how the exploitation of the GPU computing power and the readiness of the COSMO code to run on GPUs can drastically reduce the costs of such simulations.

The manuscript is well structured. The introduction is followed by a first result section focused on the computational aspects, for which the hybrid CPU/GPU setup is essential. While the main results are obvious, the discussion of the strong scaling properties of the model system would be more interesting if additional details were provided. More detailed comments follow later.

Thereafter the model simulations of the coupled ROMSOC are analyzed and compared to observations and uncoupled simulations using ROMS only. This section covers a number of aspects, but is kept a bit superficial. Where differences to ROMS are discussed, the discussion is often too general, and difference figures are missing, which makes the assessment of the differences difficult for the reader. This section needs to be improved. More detailed comments follow later.

Overall this article is worth a publication, but needs some improvement.

Detailed comments and questions

**Abstract**

L8 … ROMSOC, a newly developed regional coupled atmosphere-ocean model. …
        "newly developed" is misleading, as both model components, ROMS and COSMO, exist already, as well as the coupling software OASIS3. Further the authors cite earlier work by Byrne et al. (2016) that is also based on a coupled COSMO ROMS model. Please change the phrasing.

**1 Introduction**

L73 delete "… for the California Current System (CalCS)" and add "(CalCS)" earlier in the same sentence.

L130 … The shallow convection was switched off because this reduced our too-low cloud cover bias in the south of our model domain (section 4.5). …
        As COSMO operates on a 7 km grid, one may expect that COSMO can simulate explicit deep convection to some degree realistically. But it is unlikely that shallow convection can be simulated faithfully. This makes it likely that better results – the reduced cloud cover bias in the south of the model domain – found without parameterized shallow convection are obtained for wrong reasons. Or one would have to conclude that the shallow convection parameterization of COSMO has no skill at 7km resolution, although this parameterization was operationally used at

similar and higher resolution for NWP. Do you know why you get better results, although this is unexpected?

**2 Model configuration**

L153 … the surface net heat flux, …
= sensible and latent turbulent heat flux + net LW flux at the surface?

L157 … to COMSO … → … to COSMO …

L173 … ocean and atmosphere models use ERA5 … COSMO uses at its lateral boundaries ERA-derived … at 6-hour frequency. ROMS is forced daily at the surface outside the COSMO region …
  What is the reason for using a 6-hour frequency for COSMO and a 24-hour frequency for ROMS? Wouldn't it be more consistent to use the ERA boundary conditions at the same frequency for both model components? Further, ERA5 data are available at hourly frequency. Would it have been advantageous to drive the simulation at this higher frequency, for atmosphere and ocean?

L174 … and surface temperature, … and the snow temperature and thickness …
  Why does COSMO need surface related data from ERA? The surface of the COSMO domain is entirely over the ocean, where SST is coupled from ROMS, or land, for which COSMO can compute the surface temperature from its land model (TERRA) and the energy exchange between atmosphere and land?
  Why is ERA5 snow temperature and thickness needed for COSMO?

L178 … In addition, ROMS SST and sea surface salinity (SSS) fields are being restored to monthly Reynolds SST fields (NOAA OI v2) and climatological SSS (ICOADS, Worley et al., 2005). …
  Restoring SST seems necessary because the effect of turbulent heat fluxes at the surface on SST would otherwise be missing in the described setup of ROMS. For SSS it is less clear because surface freshwater fluxes are prescribed. Can you please provide brief justifications for the need to restore SST and SSS?

**3 Compute setup and Performance**

L185 … comprising 12 CPU (Intel Xeon E5-2690 v3 @ 2.60GHz) …
  The XC50 nodes of Piz Daint have only 1 CPU per node, each having 12 cores. Please clarify this in the text.

L191 … Figure 2b shows the strong scaling results for ROMSOC on setups ranging from 4 to 84 nodes, with 4 nodes being the lowest node number required for ROMSOC to run. The coupled model scales very well up to 64 nodes, where it reaches a parallel efficiency of 80.8%. …
  The speed-up and the implied strong scaling factor are hard to read or guess from Figure 2. For instance the "Factor of 12" indicated in panel (b) would suggest a strong scaling factor of 12/16 = 0.75 or 75%, which is less than the 80.8% stated in the text. It would be more appropriate to present the speed-up in a log-log plot, and the strong scaling factor in a separate plot with linear axis for the scaling (and still with log axis for nodes). This would be strongly appreciated.
  It would also be of interest to know the scaling of the components separately. Based on the information that ROMS is dominating the integration time, the strong scaling up to 64 nodes is probably also dictated by ROMS. But this is only a guess. This could be shown in the same or in additional panels for speed-up and strong scaling.
  The authors show in their table A1 that there exist a considerable unbalance between the compute times for COSMO and ROMS in their "ideal" 64 nodes setup. Lines [5] and [6] of Table A1 indicate about a factor 2. How is this ratio for the time integration part of both models

alone, thus after the initialization has been completed?

This unbalance could be exploited, without further total costs by making the atmospheric integration more expensive, for instance by making the domain larger (or by increasing the horizontal grid resolution or similar actions, which could improve the expected quality of the simulation. Has this been considered?

Obviously such heterogeneous setups are very sensitive to the ratio of the computational power of the CPU and GPU units on the same node. CSCS has recently started its new Alps system, and Piz Daint will be decommissioned soon. Therefore I wonder if the authors can add information on the ideal setup and scaling for a ROMSOC setup on the Alps (on an architecture of their choice). Would this unbalance change, if yes in which direction?

Such unbalances could be avoided if both models ran on GPUs. Then the resources could be distributed as needed to get a balance. But ROMS has not been porte to GPUs. Do you plan to port ROMS to GPUs to overcome this limitation? If not, is this related to missing resources or is it based on the expected difficulties for the GPU port of some components (solver for the barotropic mode)?

L213 … This efficient scaling of energy usage is also shown in Figure 2c, where the increase in energy consumption for the 64 node setup increases by only a factor of 1.3 with respect to the simulation on 4 nodes (as a result of more nodes being used), while the speed-up in runtime increases by a factor of 12 (Figure 2b). …

The energy consumption is hard to understand. More explanations are needed. The total number of operations for the model integration is the same, whatsoever the parallelization. Only the communication related to the data exchange for the parallelization increases with the number of processes. But this is usually a minor cost. Still panel (c) suggest that doubling the number of nodes from 4 to 8 and to 16 adds most strongly to the energy consumption, although the strong scaling is best at this end, meaning that both components should scale very well, meaning that the time spent per operation is nearly stable. But when increasing from 16 to 32 and to 48 nodes the energy consumption flattens, though the communication should become increasingly more expensive. Which seems to happen when increasing the node number further from 48 to 64 and to 84(?). How can this be explained?

**4 Model evaluation**

Figure 3. Please add a panel showing the differences ROMSOC – Reynolds and ROMS – Reynolds, and ROMSOC – ROMS. This would be helpful to see more directly the effect of adding the atmosphere model and the coupling to ROMS.

L242 … abormously … → … anomalously … (?)

L246 … An important reason for this increasing bias is our turning off of the SST restoring in the coupled simulation. In the uncoupled simulation, this Newtonian restoring keeps the model's SST closer to the observation, while in the coupled simulation, the model's SST is permitted to evolve freely. …

Using a Newtonian relaxation of the SST in the coupled domain would be entirely against the spirit of coupling atmosphere and ocean, as the energetic consistency would be destroyed. Therefore this explanation is not helpful. (An important reason to use the SST restoring in ROMS is that the effect of the turbulent heat fluxes on the surface temperature needs to be accounted for, somehow.) A more interesting explanation would be that the energy fluxes determining the SST are biased. And this is what is addressed to some degree in the later parts of this article. Please rephrase your explanation for the SST bias in ROMSOC.

The SST field of ROMSOC shows a cold SST bias near the northern border, for which some explanations are given. But there seems also to exist an additional bias in a narrow strip along the

north western edge (Figure 3a), which looks like an artifact from the assimilation of the model fields to the ERA5 boundary conditions. Such boundary artifacts are also evident in Figure 12. Does the framed area in Figure 3 (and in all other similar figures) include the lateral boundary regions of COSMO, where fields are modified by external boundary condition data? If so, it would be helpful to either exclude these areas (and explain this clearly) or to add a second frame that shows the internal edge of the boundary region. Then the discussion can be focused on the performance in the interior of COSMO. (And boundary problems can be discussed separately, if the authors wish to do so.)

L255 … The 10-m wind speed is also increased in the north of the domain, possibly favoring advection of cold water masses, which could lead to overall colder SST in the north of the coupling domain in ROMSOC (Figure 3). …

    Following the description ERA5 wind is used in the lateral boundary conditions. Still the 10m wind maximum at the north western edge of the ROMSOC domain is substantially stronger than the ERA5 wind, and quite similar to the independent ccmp winds. How can this be understood? Are the ERA5 winds inefficiently used for the lateral boundary conditions of COSMO? Or is this a matter of the parameterization of the 10m wind based on model level winds and surface friction, so that the 10m wind can be substantially different despite of the assimilation of ERA5 winds in the boundary region of ROMSOC atmosphere?

    It is also a kind of astonishing that the wind vectors in Fig.5b, for ERA5, show some strange small scale changes in directions along the north western edge, at ca. 47N, of the ROMSOC domain, although this panel shows a 12 year average.

    Currently the discussion is limited to the strength of the 10m wind. The discussion should also cover directional differences. For example the ROMSOC winds close to the coast are mostly tangential from the North or slightly towards off-shore. In contrast, the ERA5 10m winds are mostly westerly, thus towards the shore, where the wind is strongest. And CCMP winds are again tangential to the coast. These directional differences should matter for the coastal upwelling in ROMSOC compared to a ERA5 forced ROMS simulation.

L258 Please include the wind directions in the discussion of Figure 6a+b.

L263 … Figure 6c shows the coastal wind drop-off, …

    The data shown in this panel is unrelated to the strong-wind event on March 7th, 2011. It is a 12 year time average, as in Figure 5. Please present Figure 6c as a separate Figure, or include it in Figure 5. Then there would be less of a risk of confusion about a specific day versus a multi year average.

    Figure6c and Figure5 show time averages for the same period, and therefore Figure 6c should be a projection onto the northerly direction of the vector fields in Figure 5a+b. Figure 5a suggests a maximum northerly wind component next to the coast and a drop-off towards offshore. Figure 6c however shows a zero northerly wind next to the coast and an increase towards offshore to about -6 m/s. Is the scale of the "Alongshore wind speed" in Figure 6c correct? If yes, why is this Figure inconsistent with the vector field of Figure5a?

    A similar problem seems to exist for ERA5 vector field in Figure 5b and the alongshore wind speed in Figure 6c. Figure 5b shows essentially no northerly wind component near 40N.

L302 … The EKE averaged in a band off California (as outlined in Figure 9a) is lower in ROMSOC as compared to ROMS, although not statistically significant. …

    Beside the general low bias of ROMSOC vs. ROMS, the other difference is the bump in EKE in ROMSOC at scales of about 500 to 700 km. Do you have any speculation on the nature of this bump? Why should a 12 year average of daily EKE develop such a bump at ~500 km scale? (ROMS does not.)

L323 … However, this bias in cloud cover does not lead to enhanced incoming shortwave (SW) radiation at the surface in that region as compared to ERA5 (Figure 12a,e) and reflects the very common too-few-too-bright cloud bias known from global models (e.g., Nam et al., 2012). …

This is a too simple description of the ROMSOC – ERA5 difference. Some areas in ROMSOC, near the Southern California coast, seem to receive a stronger downward SW flux than in ERA5, while others obviously receive less. Please add a difference plot, see also the following comment.

Figure 12: Please add "ROMSOC" and "ERA" to the panel titles. Concerning the organization, please put the panels for the same variable on the same row, and add a difference plot. This would make the differences more visible.

Figure 12: The ROMSOC panels for SW, LW and the latent turbulent heat flux show boundaries along the domain edges (see earlier comment on SST). What is the reason for theses boundaries?

L328 … Turbulent surface fluxes exhibit similar patterns in ROMSOC and ERA5, … and hence increase the latent heat flux (Stevens, 2007). …

This paragraph is a bit difficult to follow because difference plots are missing. Some features described in the current text are relevant for a relatively small fraction of the area (pos. sensible heat flux). Differences in patterns are not mentioned (strongest sea-to-air sensible heat flux near south west corner of the ROMSOC domain, absent in ERA5). Finally the LW flux downward is shown but not discussed at all.

Overall the description is not accurate. The "sensible flux downward" is positive, i.e. "from the atmosphere into the ocean", in a minor area near the north western edge and at a few places on the coast only. In most of the areas the "sensible flux downward" is negative, out of the ocean to the atmosphere. Further the pattern is different. ROMSOC shows a maximum ocean-to-atmosphere sensible heat flux along in the south west along the edge. ERA5 has no such feature. Also here a difference plot would be very helpful.

**5 Discussion**

L341 … We related this cold bias to stronger wind forcing in the coupled model as compared to its uncoupled counterpart. At the same time though, this strong wind forcing leads to increased ocean mixing and a deeper and more realistically simulated MLD throughout the domain in ROMSOC as in ROMS compared to Argo observations. …

This reads as if the wind forcing allows a more realistic MLD simulation, and at the same time the SST gets to cold although the MLD is "correct". Wouldn't this simply suggest that the radiative forcing of the ocean surface temperature are a primary concern?

L362 … we want to highlight that both the spatial and temporal resolutions of COSMO are higher compared to ERA5 and that the coupling time step of ROMSOC is 144 times higher than the forcing time step applied to ROMS. Hence, we cannot identify with certainty whether the spatial or temporal resolution of the wind forcing is responsible for oceanic differences between ROMS and ROMSOC. …

This question could be addressed. For instance a sensitivity simulation where the COSMO resolution is degraded substantially could give insight into the role of horizontal resolution. Or ROMS could be forced by hourly ERA5 data instead of 24 hourly data, which would decrease the factor of the coupling interval from 144 to 6. Have such sensitivity experiments been considered?

L366 … However, we performed a sensitivity simulation for one year where we applied the coupling only daily in ROMSOC. The difference in MLD between our default ROMSOC simulation and the sensitivity run is mainly positive, indicating a 8-10 m deeper MLD in ROMSOC with a higher temporal coupling resolution …

What is the effect on SST? If the assumption is that the deeper mixing results from a higher coupling frequency, one would also expect an effect on the SST, that depends also on mixing.

L369 … Hence, we suggest that the temporal resolution of the coupling is mainly responsible for forcing stronger ocean mixing and coastal upwelling in the coupled model. …

Would you expect that a coarser resolution in COSMO, still using the same coupling frequency of the presented ROMSOC, would allow to simulate at the same quality level, despite of the increasing difficulty to represent orographic details of the coast and islands?

Further, how would a change in the ERA5 data frequency used for ROMS boundary conditions influence the quality of the ROMS simulation (MLD, upwelling)?

---

## Author Response (AR1)

Review of "ROMSOC: A regional atmosphere-ocean coupled model for
CPU-GPU hybrid system architectures" by Eirund, Leclair, Muennich, and Gruber.

General comments
This work presents the regional coupled atmosphere ocean land model ROMSOC, which is based on the existing components COSMO for the regional atmosphere and land, ROMS for the regional ocean, and OASIS3 for the coupling. While similar such models have been constructed earlier, this article points out a novel practical implementation, that is the hybrid computational setup with the ocean component running on CPUs and the atmosphere component running on GPUs of the same nodes on the computing system they have available. In their setup, where the ocean model is slower, than the atmosphere, and the billed computing costs depend only on the number of nodes and length of computation, this means that the computations for the atmosphere and land are essentially for free. Thus, the article shows how the exploitation of the GPU computing power and the readiness of the COSMO code to run on GPUs can drastically reduce the costs of such simulations.
The manuscript is well structured. The introduction is followed by a first result section focused on the computational aspects, for which the hybrid CPU/GPU setup is essential. While the main results are obvious, the discussion of the strong scaling properties of the model system would be more interesting if additional details were provided. More detailed comments follow later.
Thereafter the model simulations of the coupled ROMSOC are analyzed and compared to observations and uncoupled simulations using ROMS only. This section covers a number of aspects, but is kept a bit superficial. Where differences to ROMS are discussed, the discussion is often too general, and difference figures are missing, which makes the assessment of the differences difficult for the reader. This section needs to be improved. More detailed comments follow later.
Overall, this article is worth a publication, but needs some improvement.

We thank the reviewer very much for the many very constructive comments and suggestions. Please find our detailed answers below.

Detailed comments and questions
**Abstract**
L8 … ROMSOC, a newly developed regional coupled atmosphere-ocean model. … "newly developed" is misleading, as both model components, ROMS and COSMO, exist already, as well as the coupling software OASIS3. Further the authors cite earlier work by Byrne et al. (2016) that is also based on a coupled COSMO ROMS model. Please change the phrasing.
We agree, the text has been rephrased to "In this study, we introduce the newest model version of ROMSOC, a recently developed regional coupled atmosphere-ocean model. This new model version integrates […] ." (l. 8)

**1 Introduction**
L73 delete "… for the California Current System (CalCS)" and add "(CalCS)" earlier in the same sentence.
Done.

L130 … The shallow convection was switched off because this reduced our too-low cloud cover bias in the south of our model domain (section 4.5). …

As COSMO operates on a 7 km grid, one may expect that COSMO can simulate explicit deep convection to some degree realistically. But it is unlikely that shallow convection can be simulated faithfully. This makes it likely that better results – the reduced cloud cover bias in the south of the model domain – found without parameterized shallow convection are obtained for wrong reasons. Or one would have to conclude that the shallow convection parameterization of COSMO has no skill at 7km resolution, although this parameterization was operationally used at similar and higher resolution for NWP. Do you know why you get better results, although this is unexpected?

Yes, we agree with this point. We could imagine that the reason for the improvement in our simulation without simulated convection is due to some climate-cloud interactions as described in Hohenegger et al (2015) for sea-breeze fronts. As the clouds off southern California are mainly driven by large-scale subsidence, interacting with the cold SST at the ocean surface, they are underlying some more dynamical forcing which may be better represented without the parameterization.

**2 Model configuration**

L153 … the surface net heat flux, …
= sensible and latent turbulent heat flux + net LW flux at the surface?
Yes. We added this clarification in the text. (l. 153)

L157 … to COMSO … → … to COSMO …
Changed.

L173 … ocean and atmosphere models use ERA5 … COSMO uses at its lateral boundaries ERA-derived … at 6-hour frequency. ROMS is forced daily at the surface outside the COSMO region …
What is the reason for using a 6-hour frequency for COSMO and a 24-hour frequency for ROMS? Wouldn't it be more consistent to use the ERA boundary conditions at the same frequency for both model components? Further, ERA5 data are available at hourly frequency. Would it have been advantageous to drive the simulation at this higher frequency, for atmosphere and ocean?
We have done some sensitivity analysis using COSMO boundary forcing data also at higher frequencies and have found little influence of a higher (i.e., 3-hourly) forcing on the model output, which is why we left the forcing at 6-hourly intervals as also used before. For ROMS, there has indeed been no testing of forcing the ocean model at higher frequency. The initial reason for not using 6-hourly forcing is the simulation of the daily cycle (which becomes increasingly important for biology): ROMS would in this case linearly interpolate between the forcings to create a daily cycle which would lead to unrealistic results. At even higher frequency than 6h the data load would be too large and has hence not been considered. With the 24-hourly forcing, the daily cycle is fitted according to the location of the data, which has been proven to result in good results for simulating the daily cycle of biological and biogeochemical quantities. It would definitely we worthy testing the frequency of boundary forcing for ROMS, but then again we would lose the one to one comparison to other ocean-only model runs, which have been already performed within the group prior to our coupled runs.

L174 … and surface temperature, … and the snow temperature and thickness …
Why does COSMO need surface related data from ERA? The surface of the COSMO domain is entirely over the ocean, where SST is coupled from ROMS, or land, for which COSMO can compute the surface temperature from its land model (TERRA) and the energy exchange between atmosphere and land?
Why is ERA5 snow temperature and thickness needed for COSMO?
In fact, the COSMO domain contains a small stripe of land along the US west coast, in order to properly simulate land-sea dynamics (e.g., wind changes), see our Figure 1 in the manuscript. As this

small portion of land also contains orography, ERA5 surface data including snow properties are needed.

L178 … In addition, ROMS SST and sea surface salinity (SSS) fields are being restored to monthly Reynolds SST fields (NOAA OI v2) and climatological SSS (ICOADS, Worley et al., 2005). …

Restoring SST seems necessary because the effect of turbulent heat fluxes at the surface on SST would otherwise be missing in the described setup of ROMS. For SSS it is less clear because surface freshwater fluxes are prescribed. Can you please provide brief justifications for the need to restore SST and SSS?

The main reasons for restoring SST and SSS are to prevent model drift, improve model accuracy, compensate for limitations in the atmospheric forcing data, and enhance the skill of physical oceanic parameterizations depending on SST and SSS.

**3 Compute setup and Performance**

L185 … comprising 12 CPU (Intel Xeon E5-2690 v3 @ 2.60GHz) …

The XC50 nodes of Piz Daint have only 1 CPU per node, each having 12 cores. Please clarify this in the text.

We adapted the text accordingly. (l. 186)

L191 … Figure 2b shows the strong scaling results for ROMSOC on setups ranging from 4 to 84 nodes, with 4 nodes being the lowest node number required for ROMSOC to run. The coupled model scales very well up to 64 nodes, where it reaches a parallel efficiency of 80.8%. …

The speed-up and the implied strong scaling factor are hard to read or guess from Figure 2. For instance the "Factor of 12" indicated in panel (b) would suggest a strong scaling factor of 12/16 = 0.75 or 75%, which is less than the 80.8% stated in the text. It would be more appropriate to present the speed-up in a log-log plot, and the strong scaling factor in a separate plot with linear axis for the scaling (and still with log axis for nodes). This would be strongly appreciated.

It would also be of interest to know the scaling of the components separately. Based on the information that ROMS is dominating the integration time, the strong scaling up to 64 nodes is probably also dictated by ROMS. But this is only a guess. This could be shown in the same or in additional panels for speed-up and strong scaling.

We updated Figure 2 and now present the ROMSOC speed-up on a log-log plot. In addition, we now show parallel efficiency, which was before only mentioned in the text.

We removed the figure showing energy consumption, as this figure raised some confusion (please see our comment below).

Unfortunately, we cannot show the scaling of the components separately, as our current model setup is not (yet) running on Alps (also see some further comments below).

The authors show in their table A1 that there exist a considerable unbalance between the compute times for COSMO and ROMS in their "ideal" 64 nodes setup. Lines [5] and [6] of Table A1 indicate about a factor 2. How is this ratio for the time integration part of both models

alone, thus after the initialization has been completed?
This unbalance could be exploited, without further total costs by making the atmospheric integration more expensive, for instance by making the domain larger (or by increasing the horizontal grid resolution or similar actions, which could improve the expected quality of the simulation. Has this been considered?

Unfortunately, we did not exploit this unbalance as you suggested. As the setup of the model (defining the grid specifics, tuning, etc) was quite time intensive, we did not alter the set-up once it was defined. But we agree that in a potential next version of the model this should be exploited.

Obviously, such heterogeneous setups are very sensitive to the ratio of the computational power of the CPU and GPU units on the same node. CSCS has recently started its new Alps system, and Piz Daint will be decommissioned soon. Therefore I wonder if the authors can add information on the ideal setup and scaling for a ROMSOC setup on the Alps (on an architecture of their choice). Would this unbalance change, if yes in which direction?
Such unbalances could be avoided if both models ran on GPUs. Then the resources could be distributed as needed to get a balance. But ROMS has not been porte to GPUs. Do you plan to port ROMS to GPUs to overcome this limitation? If not, is this related to missing resources or is it based on the expected difficulties for the GPU port of some components (solver for the barotropic mode)?

This is a very good point, and we wish we could give a more satisfying answer. First, no, ROMS has not been ported to GPU. Even though ROMS is still widely used (within the group itself and elsewhere in the community), it will be replaced by ICON in the near future in our group. Hence, no resources could be put into porting ROMS to GPU at this moment. In any case though, we consider it an ideal setup as it currently is, with ROMS using the "left over" CPUs from COSMO-GPU. Unfortunately, we cannot add information of ROMSOC on Alps. As COSMO will be discontinued in the future, there was missing technical support for porting it to Alps and our current setup is (at this moment) unable to run on Alps, which we highly regret.

L213 … This efficient scaling of energy usage is also shown in Figure 2c, where the increase in energy consumption for the 64 node setup increases by only a factor of 1.3 with respect to the simulation on 4 nodes (as a result of more nodes being used), while the speed-up in runtime increases by a factor of 12 (Figure 2b). …
The energy consumption is hard to understand. More explanations are needed. The total number of operations for the model integration is the same, whatsoever the parallelization. Only the communication related to the data exchange for the parallelization increases with the number of processes. But this is usually a minor cost. Still panel (c) suggest that doubling the number of nodes from 4 to 8 and to 16 adds most strongly to the energy consumption, although the strong scaling is best at this end, meaning that both components should scale very well, meaning that the time spent per operation is nearly stable. But when increasing from 16 to 32 and to 48 nodes the energy consumption flattens, though the communication should become increasingly more expensive. Which seems to happen when increasing the node number further from 48 to 64 and to 84(?). How can this be explained?

Unfortunately, we don't know why, but we assume it's random. We moved this subplot to the appendix to avoid confusion.

**4 Model evaluation**
Figure 3. Please add a panel showing the differences ROMSOC – Reynolds and ROMS – Reynolds, and ROMSOC – ROMS. This would be helpful to see more directly the effect of adding the atmosphere model and the coupling to ROMS.

We agree, Figure 3 is updated accordingly.

L242 … abormously … → … anomalously … (?)
Changed to "anomalously".

L246 … An important reason for this increasing bias is our turning off of the SST restoring in the coupled simulation. In the uncoupled simulation, this Newtonian restoring keeps the model's SST closer to the observation, while in the coupled simulation, the model's SST is permitted to evolve freely. …
Using a Newtonian relaxation of the SST in the coupled domain would be entirely against the spirit of coupling atmosphere and ocean, as the energetic consistency would be destroyed. Therefore, this explanation is not helpful. (An important reason to use the SST restoring in ROMS is that the effect of the turbulent heat fluxes on the surface temperature needs to be accounted for, somehow.) A more interesting explanation would be that the energy fluxes determining the SST are biased. And this is what is addressed to some degree in the later parts of this article. Please rephrase your explanation for the SST bias in ROMSOC.
Thank you, we added the point of biased surface fluxes. However, we also kept the information that the SSTs are restored in the uncoupled simulation in order not to hide this important difference. It now reads "The reasons for this are twofold. Firstly, the turning off of the SST restoring in the coupled simulation permits the SST to evolve freely, as in the uncoupled simulation, this Newtonian restoring keeps the model's SST closer to the observation. Secondly, the coupled simulation exhibits biases in the surface energy fluxes (Figure 13), particularly the sensible heat flux, which is simulated too strong. This cold SST bias is particularly pronounced in the offshore region." (l. 248)

The SST field of ROMSOC shows a cold SST bias near the northern border, for which some explanations are given. But there seems also to exist an additional bias in a narrow strip along the north western edge (Figure 3a), which looks like an artifact from the assimilation of the model fields to the ERA5 boundary conditions. Such boundary artifacts are also evident in Figure 12. Does the framed area in Figure 3 (and in all other similar figures) include the lateral boundary regions of COSMO, where fields are modified by external boundary condition data? If so, it would be helpful to either exclude these areas (and explain this clearly) or to add a second frame that shows the internal edge of the boundary region. Then the discussion can be focused on the performance in the interior of COSMO. (And boundary problems can be discussed separately, if the authors wish to do so.)
Yes, these boundaries are included, as the panel shows the whole COSMO domain and hence these boundary effects are shown. We added a second frame in Figure 3a which excludes the boundary of the domain where the ERA5 boundary conditions are assimilated. There are still boundary effects seen further inside the domain, but as these boundary regions are not included in the analyses, we consider these effects to be less relevant for our study.
We included a note on the boundary problem in the manuscript ("Also note the boundary artifacts especially along the northwestern edge of the COSMO domain, which arise from assimilating the ERA5 boundary conditions. Even though we allow for a transition layer of 7 grid points along the COSMO edge (see Section 2.3), boundary effects are still visible along these edges. As these regions are however not included in any analyses, we do not investigate these boundary problems further.", l. 234).

L255 … The 10-m wind speed is also increased in the north of the domain, possibly favoring advection of cold water masses, which could lead to overall colder SST in the north of the coupling domain in ROMSOC (Figure 3). …
Following the description ERA5 wind is used in the lateral boundary conditions. Still the 10m wind maximum at the north western edge of the ROMSOC domain is substantially stronger than the ERA5 wind, and quite similar to the independent ccmp winds. How can this be understood? Are the ERA5 winds inefficiently used for the lateral boundary conditions of COSMO? Or is this a matter of the parameterization of the 10m wind based on model level winds and surface friction, so that the 10m

wind can be substantially different despite of the assimilation of ERA5 winds in the boundary region of ROMSOC atmosphere?

Yes, we also assume that this is due to the parameterization of the 10m wind within the model.

It is also a kind of astonishing that the wind vectors in Fig.5b, for ERA5, show some strange small scale changes in directions along the north western edge, at ca. 47N, of the ROMSOC domain, although this panel shows a 12 year average.

There was an error in plotting the wind vectors, this is now fixed (see new Figure 5). We thank the reviewer for pointing this out.

Currently the discussion is limited to the strength of the 10m wind. The discussion should also cover directional differences. For example the ROMSOC winds close to the coast are mostly tangential from the North or slightly towards off-shore. In contrast, the ERA5 10m winds are mostly westerly, thus towards the shore, where the wind is strongest. And CCMP winds are again tangential to the coast. These directional differences should matter for the coastal upwelling in ROMSOC compared to a ERA5 forced ROMS simulation.

As pointed out above, there was an error in plotting the ERA5 wind vectors. This is now corrected. We anyways added to the discussion of the wind vectors (see Section 4.2).

L258 Please include the wind directions in the discussion of Figure 6a+b.

We added the wind directions to Figure 6 and added some comments on this in the discussion (see Section 4.2).

L263 … Figure 6c shows the coastal wind drop-off, …

The data shown in this panel is unrelated to the strong-wind event on March 7th, 2011. It is a 12 year time average, as in Figure 5. Please present Figure 6c as a separate Figure, or include it in Figure 5. Then there would be less of a risk of confusion about a specific day versus a multi year average.

We agree and now present Figure 6c as a separate Figure (new Figure 7).

Figure6c and Figure5 show time averages for the same period, and therefore Figure 6c should be a projection onto the northerly direction of the vector fields in Figure 5a+b. Figure 5a suggests a maximum northerly wind component next to the coast and a drop-off towards offshore. Figure 6c however shows a zero northerly wind next to the coast and an increase towards offshore to about -6 m/s. Is the scale of the "Alongshore wind speed" in Figure 6c correct? If yes, why is this Figure inconsistent with the vector field of Figure5a?

Yes, the wind speed along the shore (hence the northerly component) should decrease, this is correct. In Figure 5a however, we thinned out the vector field for plotting, as otherwise the plot would be too busy and the underlying colours would not be visible. Hence, not every arrow (also the ones right next to the coast) are plotted and this drop is not visible with this visualization. As we preferred to keep the thinned out wind field for the vectors, we added this information in the figure caption to not confuse the reader with this discrepancy ("Note that for visualization purposes, the vector wind field has been thinned out and not every grid point is plotted.").

A similar problem seems to exist for ERA5 vector field in Figure 5b and the alongshore wind speed in Figure 6c. Figure 5b shows essentially no northerly wind component near 40N.

As pointed out above, there was an error in the plotting of the vector field, which is now fixed.

L302 … The EKE averaged in a band off California (as outlined in Figure 9a) is lower in ROMSOC as compared to ROMS, although not statistically significant. …

Beside the general low bias of ROMSOC vs. ROMS, the other difference is the bump in EKE in ROMSOC at scales of about 500 to 700 km. Do you have any speculation on the nature of this bump? Why should a 12 year average of daily EKE develop such a bump at ~500 km scale?
(ROMS does not.)
We thank the reviewer for pointing this out and agree that this bump is a striking feature. We unfortunately cannot give a definite answer, but one speculation could be that it arises from the colder SST in the upwelling region in ROMSOC and thus a sharper transition between the coastal and open ocean zone, which would match the scale of this bump.
But as it is not statistically significant, we don't want to conclusively decide on one reason, as it might also simply be random.

Figure 12: Please add "ROMSOC" and "ERA" to the panel titles. Concerning the organization, please put the panels for the same variable on the same row, and add a difference plot. This would make the differences more visible.
We included all suggestions in the new Figure 13 (previously Figure 12).

Figure 12: The ROMSOC panels for SW, LW and the latent turbulent heat flux show boundaries along the domain edges (see earlier comment on SST). What is the reason for theses boundaries?
These are the effects of the COSMO model boundaries, where COSMO interpolates between the modelled variables and the boundary forcing. Hence, these boundary effects are not physical, but rather numerical. We added a note on these boundary effects in Section 4.1.

L323 … However, this bias in cloud cover does not lead to enhanced incoming shortwave (SW) radiation at the surface in that region as compared to ERA5 (Figure 12a,e) and reflects the very common too-few-too-bright cloud bias known from global models (e.g., Nam et al., 2012). …
This is a too simple description of the ROMSOC – ERA5 difference. Some areas in ROMSOC, near the Southern California coast, seem to receive a stronger downward SW flux than in ERA5, while others obviously receive less. Please add a difference plot, see also the following comment.
We agree that our discussion was too simple. We updated the figure according to the suggestions and added to the text: "This bias is reflected in biases in the surface radiative fluxes in ROMSOC: the model features enhanced incoming shortwave (SW) radiation of up to 38 W m-2 at the surface in that region as compared to ERA5 (Figure 13 a,b,c), resulting in the warm SST bias of 2C seen in Figure 3. This warmer surface in turn increases the outgoing LW radiation in ROMSOC up to 28 W m-2. The northern and western parts of the domain are characterized by reduced incoming SW and outgoing LW radiation in ROMSOC (Figure 13 c,f), contributing to the cold SST bias offshore (Figure 3). This bias in radiative fluxes can be attributed to too high cloud cover in the south/southwest of the domain and possibly too bright clouds in the north, as cloudfraction in ROMSOC is well-represented in that region." (l. 327).

L328 … Turbulent surface fluxes exhibit similar patterns in ROMSOC and ERA5, … and hence increase the latent heat flux (Stevens, 2007). …
This paragraph is a bit difficult to follow because difference plots are missing. Some features described in the current text are relevant for a relatively small fraction of the area (pos. sensible heat flux). Differences in patterns are not mentioned (strongest sea-to-air sensible heat flux near south west corner of the ROMSOC domain, absent in ERA5). Finally the LW flux downward is shown but not discussed at all.
Overall the description is not accurate. The "sensible flux downward" is positive, i.e. "from the atmosphere into the ocean", in a minor area near the north western edge and at a few places on the coast only. In most of the areas the "sensible flux downward" is negative, out of the ocean to the

atmosphere. Further the pattern is different. ROMSOC shows a maximum ocean-to-atmosphere sensible heat flux along in the south west along the edge. ERA5 has no such feature. Also here a difference plot would be very helpful.

We added the difference plot to Figure 13 (previously Figure 12).
In addition, we rewrote the discussion of cloud patterns and radiative fluxes (see Section 4.5).

**5 Discussion**

L341 … We related this cold bias to stronger wind forcing in the coupled model as compared to its uncoupled counterpart. At the same time though, this strong wind forcing leads to increased ocean mixing and a deeper and more realistically simulated MLD throughout the domain in ROMSOC as in ROMS compared to Argo observations. …

This reads as if the wind forcing allows a more realistic MLD simulation, and at the same time the SST gets too cold although the MLD is "correct". Wouldn't this simply suggest that the radiative forcing of the ocean surface temperature are a primary concern?

We agree that biases in radiative forcing are also responsible for the SST biases and added this so the discussion: "…We relate this cold bias to biases in the radiative surface forcing in the model (reduced incoming SW radiation and an increased upward sensible heat flux in offshore regions) and in addition to stronger wind forcing in the coupled model as compared to its uncoupled counterpart. " (l. 357)

L362 … we want to highlight that both the spatial and temporal resolutions of COSMO are higher compared to ERA5 and that the coupling time step of ROMSOC is 144 times higher than the forcing time step applied to ROMS. Hence, we cannot identify with certainty whether the spatial or temporal resolution of the wind forcing is responsible for oceanic differences between ROMS and ROMSOC. …

This question could be addressed. For instance a sensitivity simulation where the COSMO resolution is degraded substantially could give insight into the role of horizontal resolution. Or ROMS could be forced by hourly ERA5 data instead of 24 hourly data, which would decrease the factor of the coupling interval from 144 to 6. Have such sensitivity experiments been considered?

We agree that this would be an interesting sensitivity study. However, as Piz Daint, where we ran our coupled simulations, is no longer existing, we unfortunately cannot perform such simulations at this point.
Also, for ROMS, such simulations have not been performed, but could definitely be part of a future study, especially when analyzing e.g., the daily cycle of biological quantities.

We added "This sensitivity to forcing frequency should be investigated in more details in future studies, where for example also ROMS could be forced by hourly ERA5 data instead of daily data. Such an experiment could in addition be used to analyze daily cycles in marine biology and its sensitivity to atmospheric forcing." (l. 381) to the text, to point out this potential for future work.

L366 … However, we performed a sensitivity simulation for one year where we applied the coupling only daily in ROMSOC. The difference in MLD between our default ROMSOC simulation and the sensitivity run is mainly positive, indicating a 8-10 m deeper MLD in ROMSOC with a higher temporal coupling resolution …
L369 … Hence, we suggest that the temporal resolution of the coupling is mainly responsible for forcing stronger ocean mixing and coastal upwelling in the coupled model. …

Would you expect that a coarser resolution in COSMO, still using the same coupling frequency of the presented ROMSOC, would allow to simulate at the same quality level, despite of the increasing difficulty to represent orographic details of the coast and islands?
Further, how would a change in the ERA5 data frequency used for ROMS boundary conditions influence the quality of the ROMS simulation (MLD, upwelling)?

Regarding the COSMO resolution: no, we would not expect the same quality level. We had a coarser resolution in some early versions of the model and the results were poorer, due to even larger biases in cloud cover and radiative forcing and also wind strength.

We agree that changing the ERA5 data forcing frequency for ROMS remains an interesting point that should be tested. However, even an hourly forcing of ROMS with ERA5 would still not include the wind drop off and not include the short-term and small-scale effects of peaks in wind speed. Hence, we don't expect the same level of model-skill from a more frequently forced ROMS simulation as compared to a coupled ROMSOC simulation.

What is the effect on SST? If the assumption is that the deeper mixing results from a higher coupling frequency, one would also expect an effect on the SST, that depends also on mixing?
Yes, we agree, that enhanced mixing with a higher coupling frequency results in a colder SST. However, as pointed out as well, SST is also highly dependent on surface radiative fluxes.

This manuscript proposes a novel regional coupled atmosphere-ocean model (ROMSOC) that integrates COSMO and ROMS while optimizing performance on a CPU-GPU hybrid architecture. The model is applied for over Northeast Pacific with a focus on the California Current System, evaluating key air-sea interactions over a 12-year hindcast period. The manuscript also provides a detailed discussion of the potential sources of uncertainties for SST simulation bias. In general, the study presents a interesting model development work with a insightful scientific study on SST with solid method. The manuscript is well organized with detailed description of the model and thorough discussions of the result. Therefore, I would recommend it to be accepted if the following comments could be properly addressed.

We thank the reviewer very much for the constructive comments, questions and suggestions. Please find our detailed answers below.

Comment#1: Not really a comment but more of a curious question: the study demonstrates impressive computational efficiency with effective energy cost on a heterogeneous CPU-GPU system, but for a fixed grid configuration. How does the performance scale with different grid sizes? Similarly, as the author mentioned at line#203 it is constrained by the CPU running ROMS part, how does the performance scale with more CPU nodes? Moreover, although the practice seems very promising, is there any potential limitations, for example, is it applicable for ultra-find grid resolution like 1km?

We did some sensitivity experiments regarding different grid sizes in the beginning of the set-up process of our coupled model and we found our current setup to be the most efficient. As other model specifics (tuning parameter, parameterizations) have changed since these experiments, we did not include these run times in the final benchmarking analysis for this publication. The ROMS model grid was somewhat set fixed, as we compared the model output of our coupled model to model simulations of the ocean-only model, which was already set up on the specified grid.

Similarly, for the pure CPU version we did not benchmark the model, as we did not plan to run the model on CPU in production mode.

Yes, theoretically, the model could be run on grid resolutions as low as 1 km. This would of course increase the model run time substantially, especially also because the time stepping frequency of COSMO had to be increased as well. Hence, we did not test this simulation for our purposes, but it would definitely be interesting to investigate.

Comment#2: The manuscript generally presents very detailed discussions regarding the potential causes for simulation bias by the coupled model, but there are limited discussions to quantify the uncertainties induced or reduced by the coupling effort. For example, SST bias of ROMSOC is primarily attributed to wind bias which indicated that the raw SST bias of ROMS is amplified by COSMO. But there is no demonstration of the 10m-wind bias by pure COSMO. I would recommend the revision to consider incorporate such kind of comparison and discussion to help better demonstrate the benefit and "side-effect" of this coupled model.

We agree that the manuscript would benefit from such a discussion. However, as our focus was the ocean response to a coupled/non-coupled atmosphere, we focused our analyses and model simulations on this issue and unfortunately did not run an atmosphere-only simulation to compare. We only performed one-year simulations of COSMO to access sensitivities of tuning parameters, grid size etc., but did not simulate beyond that point. As we were always constrained by compute time, we unfortunately could not run several longer COSMO-only simulations. As Piz Daint (where we ran our coupled model) is now replaced by its follow-up system Alps and our model is currently not (yet) set up to run on Alps, it is at this moment impossible for us to run such a longer COSMO-only simulation, which we highly regret.

Nevertheless, we expanded the discussion of the SST bias and also now relate this to biases in the radiative fluxes (new Figure 13 and Sections 4.5 and 5).

Comment#3: The study acknowledges a common bias in climate models as there are too few clouds in the southern domain over CalCS region. A more detailed discussion on why ROMSOC exhibits this behavior would be helpful to understand the limit of this specific model. This may not be directly falls into the concern of this study but: some double-momentum mechanisms have been developed for years, does COSMO support any of them, or there is any study ever try it?

We agree and added to the discussion on the cloud bias in ROMSOC ("Reasons for this bias are manifold. Firstly, an inadequate representation of boundary layer as the height and extent of the temperature inversion and turbulent processes within the boundary layer can cause a misrepresentation of low-lying clouds (Brient et al., 2019, Heim et al., 2021). However, even on our kilometer-scale resolution, these processes are not yet adequately resolved and one would need vertical resolutions down to the meter-scale, which is unfortunately not computationally possible for our setup. In addition, cloud and turbulence parameterizations still may oversimplify the complex subgrid-scale processes within clouds and lead to an underestimation of cloud cover (Brient et al., 2019). Thirdly, ROMSOC exhibits a warm SST bias (see new Figure xx) in the south of our modeling domain. These warm SSTs reduce the stability of the lower atmosphere, which inhibits stratocumulus cloud formation and maintenance (Lin et al., 2014).") (l. 394)

To summarize our addition, we suspect the main reason for the too-few cloud bias to be an inadequate representation of boundary layer processes (Brient et al., 2019, Heim et al., 2021). As the low-lying stratocumulus clouds present in this region form within the boundary layer, the resolution of the temperature inversion and turbulent processes is essential. However, even on our kilometer-scale resolution, these processes are not yet adequately resolved and one would need vertical resolutions down to the meter-scale, which is unfortunately not computationally possible for our setup. In addition, cloud and turbulence parameterizations still may oversimplify the complex subgrid-scale processes within clouds and lead to an underestimation of cloud cover (Brient et al., 2019). Thirdly, ROMSOC exhibits a warm SST bias (see new Figure xx) in the south of our modeling domain. These warm SSTs reduce the stability of the lower atmosphere, which inhibits stratocumulus cloud formation and maintenance (Lin et al., 2014).

To add to the COSMO 2-M scheme implementation, we agree, that it would have been of added value to test the impact of such a more advanced parameterization and access, if the potential improvement is worth the additional computational cost. The COSMO-CPU version does support a 2-M cloud microphysics scheme and it has been applied/tested against the 1-M scheme previously (e.g., Eirund et al., 2021).

However, the COSMO 2-M scheme has not been ported to GPU, hence we could not test it in the COSMO-GPU version. As marine low-lying stratocumulus clouds are however primarily driven by

surface forcing and boundary layer processes, we suspect the additional impact of a 2-M scheme to be small in comparison to e.g., higher vertical resolution.